# The IRE1α/XBP1 signaling axis drives myoblast fusion in adult skeletal muscle

Aniket S Joshi [1,4], Meiricris Tomaz da Silva [1,4], Anirban Roy [1,4], Tatiana E Koike[1], Mingfu Wu[1], Micah B Castillo[2], Preethi H Gunaratne[2], Yu Liu[2], Takao Iwawaki [3] & Ashok Kumar [1✉]

## Abstract

Skeletal muscle regeneration involves a signaling network that regulates the proliferation, differentiation, and fusion of muscle precursor cells to injured myofibers. IRE1α, one of the arms of the unfolded protein response, regulates cellular proteostasis in response to ER stress. Here, we demonstrate that inducible deletion of IRE1α in satellite cells of mice impairs skeletal muscle regeneration through inhibiting myoblast fusion. Knockdown of IRE1α or its downstream target, X-box protein 1 (XBP1), also inhibits myoblast fusion during myogenesis. Transcriptome analysis revealed that knockdown of IRE1α or XBP1 dysregulates the gene expression of molecules involved in myoblast fusion. The IRE1α-XBP1 axis mediates the gene expression of multiple profusion molecules, including myomaker (*Mymk*). Spliced XBP1 (sXBP1) transcription factor binds to the promoter of *Mymk* gene during myogenesis. Overexpression of myomaker in IRE1α-knockdown cultures rescues fusion defects. Inducible deletion of IRE1α in satellite cells also inhibits myoblast fusion and myofiber hypertrophy in response to functional overload. Collectively, our study demonstrates that IRE1α promotes myoblast fusion through sXBP1-mediated up-regulation of the gene expression of multiple profusion molecules, including myomaker.

Key words Muscle Regeneration; IRE1; XBP1; Myoblast Fusion; and Myomaker
Subject Categories Membranes & Trafficking; Musculoskeletal System; Signal Transduction

## Introduction

Skeletal muscle cells, more commonly known as myofibers, are multinucleated syncytia that arise by the fusion of thousands of myoblasts during development (Buckingham et al, 2003). Adult skeletal muscle tissue retains regenerative capability mainly due to the presence of a pool of muscle progenitor cells, known as satellite cells, which reside in basal lamina around myofibers in a quiescent state (Yin et al, 2013). Following muscle injury, satellite cells are activated, which then proliferate, differentiate into myoblasts, and form multinucleated myotubes through the fusion of myoblast with another myoblast or nascent myotubes. Moreover, a subset of myoblasts also fuses with damaged pre-existing myofibers to accomplish muscle repair (Relaix et al, 2021; Yin et al, 2013).

Myoblast fusion is a systematic process that involves migration and alignment of membranes of fusion-competent myoblasts, remodeling of the cytoskeleton at contact sites followed by the opening of fusion pores to allow movement of cytoplasmic content, and eventually amalgamation of two myogenic cells into one (Abmayr and Pavlath, 2012; Hochreiter-Hufford et al, 2013; Kim et al, 2015). Recently, two transmembrane proteins, named myomaker and myomerger (also known as myomixer and minion) have been identified as major drivers of myoblast fusion in diverse conditions (Bi et al, 2017; Millay et al, 2013; Millay et al, 2014; Zhang et al, 2020). It is now increasingly clear that myoblast fusion is regulated by multiple signaling pathways that are activated due to the interaction of specific membrane proteins between fusion partners or as a part of myogenic differentiation program (Krauss, 2010). However, the molecular and signaling mechanisms regulating myoblast fusion remain less understood.

Skeletal muscle regeneration requires the synthesis and processing of many proteins, including growth factors, signaling proteins, and a new set of contractile, cytoskeletal, and membrane proteins (Yin et al, 2013). The endoplasmic reticulum (ER) is the site for the folding of membrane and secretory proteins, synthesis of lipids and sterols, and storage of free calcium. Physiological demands or pathological stresses, such as the presence of mutated proteins that cannot properly fold in the ER, can lead to the accumulation of unfolded protein, thereby causing ER stress (Adams et al, 2019). Mammalian cells respond to the presence of unfolded or misfolded proteins within the ER through the activation of an intricate set of signal pathways, collectively termed the unfolded protein response (UPR) that improves ER folding capacity and restores cellular homeostasis. The UPR consists of three ER-resident proteins: inositol requiring enzyme 1α/β (IRE1), PKR-like ER kinase (PERK), and activating transcription factor 6α/β (ATF6) (Wang and Kaufman, 2014; Wu and Kaufman, 2006). Among these, the IRE1 is the most ancient and conserved branch of the UPR that plays a major role in resolving ER stress. The cytosolic portion of IRE1 possesses a kinase domain that

[1]Department of Pharmacological and Pharmaceutical Sciences, University of Houston College of Pharmacy, Houston, TX 77204, USA. [2]Department of Biology and Biochemistry, University of Houston, Houston, TX 77204, USA. [3]Division of Cell Medicine, Department of Life Science, Medical Research Institute, Kanazawa Medical University, Uchinada, Japan. [4]These authors contributed equally: Aniket S Joshi, Meiricris Tomaz da Silva, Anirban Roy. ✉E-mail: akumar43@Central.UH.EDU

autophosphorylates, leading to the stimulation of its endoribonuclease activity, which catalyzes unconventional processing of the mRNA encoding X-box-binding protein 1 (XBP1) and creates a transcriptionally active XBP1, known as spliced XBP1 (sXBP1). This results in the stimulation of gene expression of many molecules that increase the protein-folding capacity as well as augment protein degradation and transport pathways, which mitigates the burden of misfolded protein within the ER. IRE1 activation can also lead to promiscuous endoribonuclease activity that causes mRNA decay at the ER membrane, a process called regulated IRE1-dependent decay (RIDD), thus reducing the protein load. Finally, during ER stress, the kinase domain of IRE1 binds to the TRAF2 adapter protein that results in the phosphorylation and activation of JNK to control cell fate (Acosta-Alvear et al, 2018; Wang and Kaufman, 2014; Wu and Kaufman, 2006). Remarkably, signaling pathways activated by stimulation of various cell surface receptors have been found to crosstalk with UPR signaling in an ER stress-independent manner suggesting that ER sensors also mediate non-canonical UPR responses to regulate cell physiology (Hetz, 2012; Hetz et al, 2020).

Accumulating evidence suggests that UPR pathways play important roles in skeletal muscle development and regenerative myogenesis (Afroze and Kumar, 2019; Bohnert et al, 2018). Pharmacological activation of ER stress has been found to augment myotube formation in cultured myoblasts following induction of differentiation (Nakanishi et al, 2007). Recent studies have also demonstrated that the PERK arm of the UPR is essential for self-renewal of satellite cells (Xiong et al, 2017; Zismanov et al, 2016). We have recently demonstrated that IRE1α signaling in myofibers promotes skeletal muscle regeneration following acute injury as well as in the mdx model of Duchenne muscular dystrophy (Roy et al, 2021). However, the cell-autonomous role of IRE1α in the regulation of satellite cell function in adult skeletal muscle remained completely unknown.

In the present study, using genetic and molecular approaches, we demonstrate that satellite cell-specific ablation of IRE1α (gene name: Ern1) impairs skeletal muscle regeneration in adult mice. Importantly, IRE1α does not influence satellite cell abundance or differentiation in regenerating skeletal muscle. Rather, IRE1α promotes muscle repair and growth through augmenting myoblast fusion. Overexpression of IRE1α or its downstream target sXBP1 in cultured myoblasts results in the formation of myotubes having an increased diameter following induction of differentiation. The IRE1α/XBP1 signaling augments the gene expression of multiple profusion molecules, including myomaker in differentiating myoblasts. Finally, our results demonstrate that IRE1α signaling in muscle progenitor cells is also essential for overload-induced myofiber hypertrophy in adult mice.

# Results

## Activation of IRE1α in muscle progenitor cells

Pax7 transcription factor is a widely used marker expressed in both quiescent and activated satellite cells (Seale et al, 2000). By performing an immunofluorescence assay, we first examined whether the IRE1α protein is activated in satellite cells. TA muscle of wild-type mice was injured by intramuscular injection of 1.2% BaCl$_2$ solution. Results showed that p-IRE1α co-localized with Pax7 protein in both uninjured and injured TA muscle, suggesting that IRE1α is activated in both

quiescent and activated satellite cells in vivo (Fig. 1A). Furthermore, the number of p-IRE1$^+$/Pax7$^+$ cells were significantly increased in 5d-injured TA muscle compared to corresponding uninjured muscle (Fig. 1B). We next studied the activation of IRE1α in cultured myogenic cells. Mouse primary myoblasts incubated in growth medium (GM) were immunostained for Pax7, and p-IRE1α or total IRE1α protein. Consistent with in vivo results, we found that both p-IRE1α and IRE1α protein were present in Pax7$^+$ cells (Fig. 1C). Following a FACS-based intracellular protein detection assay, we also investigated whether p-IRE1α protein is present in cultured muscle progenitor cells. Mouse primary myoblasts were collected and analysed by FACS for α7-Integrin and p-IRE1α or IRE1α protein. Results showed that p-IRE1α and total IRE1α protein were present in the α7-Integrin$^+$ satellite cells (Fig. 1D). We next sought to determine how the levels of p-IRE1α and total IRE1α protein are regulated during myogenic differentiation. Primary myoblasts were incubated in differentiation medium (DM) for 0, 6, 12, 24, 48, and 72 h and the cell extracts made were analysed by western blot. Results showed that p-IRE1α and total IRE1α proteins, along with Pax7 protein, were highly abundant in myoblast cultures incubated in GM. However, the levels of p-IRE1α, IRE1α, and Pax7 protein were gradually reduced whereas the muscle differentiation markers, myogenin and myosin heavy chain (MyHC) were increased after incubation in DM (Fig. 1E). In response to ER stress, IRE1α gets activated which causes unconventional splicing of XBP1 mRNA resulting in the formation of spliced XBP1 (sXBP1) mRNA (Yoshida et al, 2001). Western blot analysis showed a peak increase in the protein levels of sXBP1 at 24 h of the addition of DM (Fig. 1E). Altogether, these results suggest that the IRE1α/XBP1 arm of the UPR is activated in muscle progenitor cells both in vivo and in vitro.

## Ablation of IRE1α in satellite cells inhibits skeletal muscle regeneration

We next investigated whether IRE1α (gene name: Ern1) signaling in satellite cells affects skeletal muscle regeneration in adult mice. Floxed Ern1 mice (henceforth Ern1$^{fl/fl}$) were crossed with tamoxifen-inducible satellite cell-specific Cre mice (Pax7-CreERT2) to generate Pax7-CreERT2;Ern1$^{fl/fl}$ (henceforth, Ern1$^{scKO}$) and littermate Ern1$^{fl/fl}$ mice (Fig. 2A). Eight-week-old Ern1$^{scKO}$ mice were given intraperitoneal injection of tamoxifen for 4 consecutive days. The mice were maintained on tamoxifen-containing chow for the entire duration of the study to ensure IRE1α deletion. Ern1$^{fl/fl}$ mice were also subjected to the same tamoxifen regimen and served as controls. One side TA muscle was injured by intramuscular injection of 1.2% BaCl$_2$ solution whereas the contralateral muscle served as uninjured control. The TA muscle was isolated at day 5 or 14 post-injury (Fig. 2B). Interestingly, the wet weight of 5d-injured TA muscle normalized by the body weight of Ern1$^{scKO}$ mice was significantly reduced compared to Ern1$^{fl/fl}$ mice (Fig. 2C–E). Next, transverse sections of uninjured and injured TA muscle were generated and analyzed by performing H&E staining (Fig. 2F). Results showed that the average myofiber cross-sectional area (CSA) was significantly reduced in injured TA muscle of Ern1$^{scKO}$ mice compared to the corresponding injured TA muscle of Ern1$^{fl/fl}$ mice both at 5d and 14d post-injury (Fig. 2F–H). To confirm the inactivation of IRE1α in satellite cells of Ern1$^{scKO}$ mice, we performed double immunostaining for p-IRE1α and Pax7 protein and observed that the proportion of p-IRE1$^+$/Pax7$^+$ cells was drastically reduced in 5d-injured TA muscle of Ern1$^{scKO}$ mice compared to the corresponding muscle of Ern1$^{fl/fl}$ mice (Fig. EV1A).

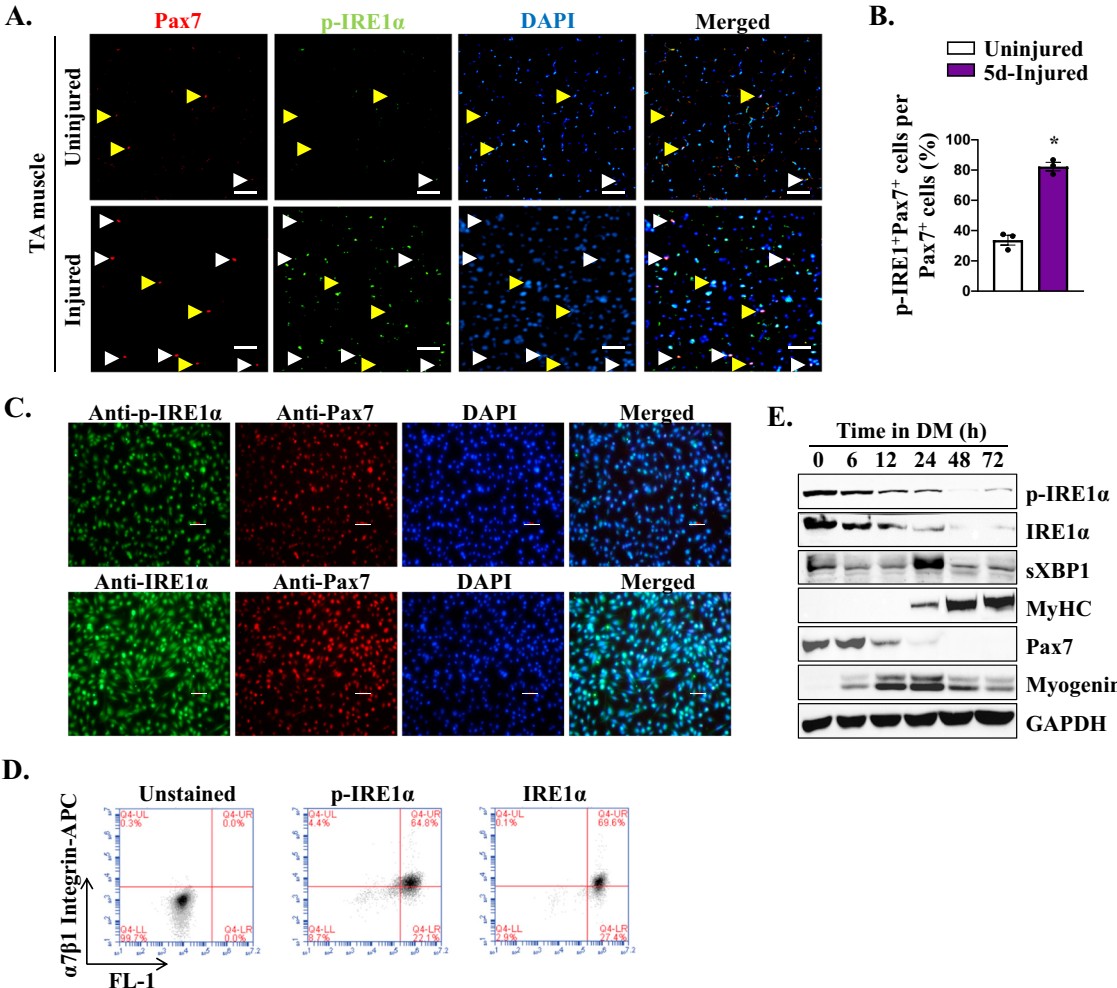

**Figure 1. Activation of IRE1α in muscle progenitor cells.**

(A) Representative photomicrographs of uninjured and 5d-injured TA muscle of wild-type mice immunostained for p-IRE1α and Pax7 protein and DAPI staining. Yellow arrowheads point to Pax7 positive cells whereas white arrowheads point to Pax7 and p-IRE1α double-positive cells. Scale bar: 100 μm. (B) Quantification of the proportion of p-IRE1 and Pax7 double-positive cells per Pax7+ cells in uninjured and 5d-injured TA muscle of mice. $n = 3$ mice in each group. Data information: All data were presented as mean ± SEM and analyzed by unpaired Student t-test. *$p \leq 0.05$, values significantly different from uninjured muscle. (C) Representative photomicrographs of wild-type primary myoblasts after immunostaining for p-IRE1α or IRE1α and Pax7 protein and DAPI staining. Scale bar: 50 μm. (D) Representative scatter plots of FACS-based analysis demonstrate the presence of p-IRE1+ and IRE1+ cells amongst the α7-Integrin+ population. $n = 3$ (biological replicates) in each group. (E) Immunoblots presented here demonstrate the relative levels of p-IRE1α, IRE1α, sXBP1, MyHC, Pax7, myogenin, and an unrelated protein GAPDH, in primary myoblasts after incubation in DM for indicated time. Source data are available online for this figure.

We also investigated the effect of satellite cell-specific deletion of IRE1α on the levels of IRE1α and other ER stress markers in the skeletal muscle of mice. Western blot analysis showed that the levels of p-IRE1α, total IRE1α, and to some extent sXBP1 were reduced whereas there was no difference in the levels of total PERK and CHOP in 5d-injured TA muscle of Ern1scKO mice compared to corresponding 5d-injured TA muscle of Ern1fl/fl mice (Fig. EV1B). These results suggest that satellite cell-specific deletion of IRE1α inhibits muscle regeneration in adult mice.

## Targeted ablation of IRE1α inhibits the growth of newly formed myofibers

Skeletal muscle regeneration is dependent on the hierarchical expression of myogenic regulatory factors (MRFs, i.e., Myf5, MyoD, and myogenin) and the early regeneration marker, an embryonic

isoform of myosin heavy chain (eMyHC), which ultimately determines the efficiency of muscle repair (Beauchamp et al, 2000; Kuang et al, 2007). We next investigated the effect of IRE1α deletion in satellite cells on the levels of eMyHC and MRFs in the skeletal muscle of mice. Transverse sections of uninjured and 5d-injured TA muscle of Ern1fl/fl and Ern1scKO mice were immunostained for eMyHC and laminin (to label boundaries of myofibers) protein. There was no significant difference in the percentage of eMyHC+ myofibers per laminin-stained cells in 5d-injured TA muscle of Ern1scKO mice compared to the corresponding muscle of Ern1fl/fl mice (Fig. 3A,B). However, average CSA and percentage of eMyHC+ myofibers containing 2 or more centrally located nuclei were significantly reduced in 5d-injured TA muscle of Ern1scKO mice compared to Ern1fl/fl mice (Fig. 3C,D). Western blot analysis showed that protein levels of MyoD and myogenin were

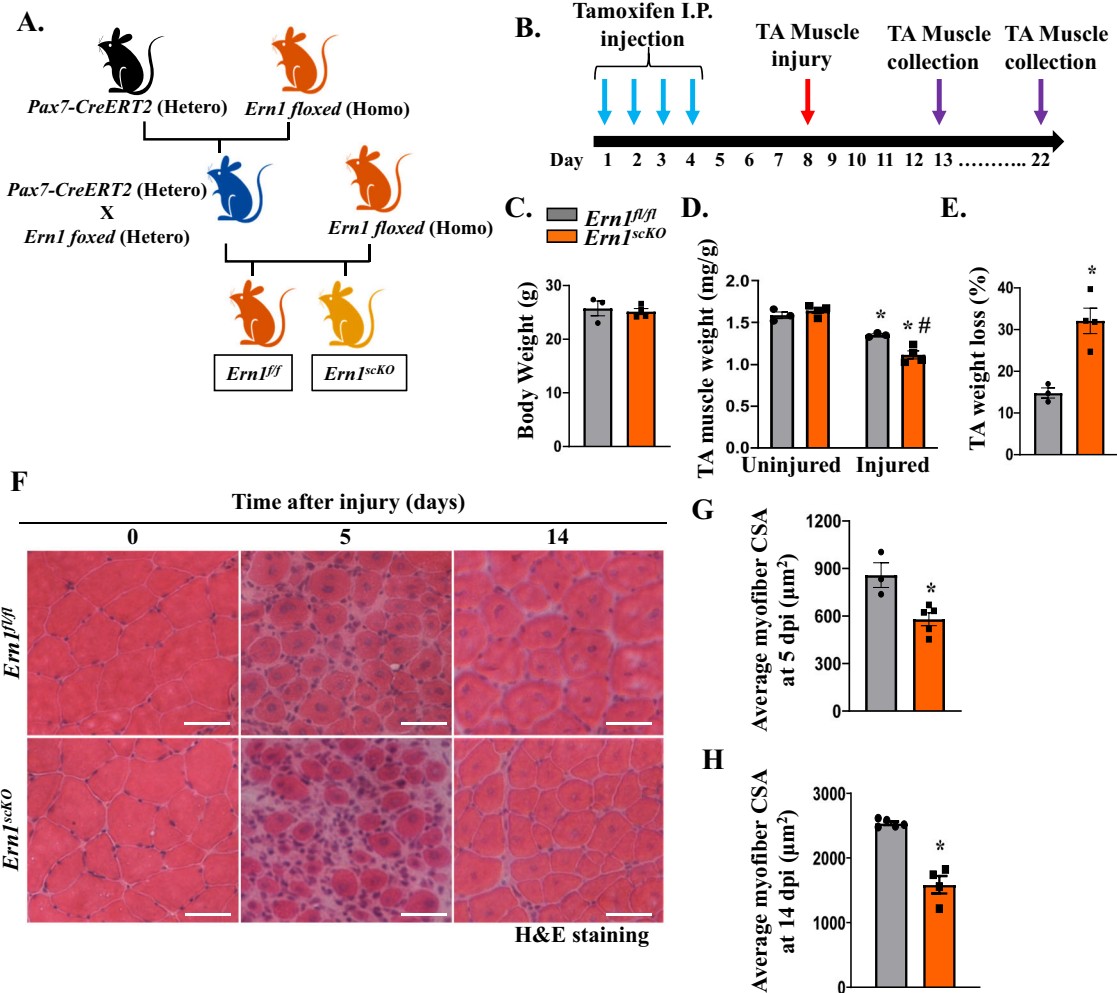

**Figure 2. Satellite cell-specific ablation of IRE1α impairs skeletal muscle regeneration.**

(A) Schematic representation of the breeding strategy used for the generation of *Ern1fl/fl* and *Ern1scKO* mice. (B) Schematic representation of the experimental design indicating the time of *Ern1* deletion, muscle injury, and collection. (C) Body weight of *Ern1fl/fl* ($n = 3$) and *Ern1scKO* ($n = 4$) mice at day 5 after intramuscular injection of 1.2% $BaCl_2$ in TA muscle. Data information: Data were presented as mean ± SEM. No significance was observed by unpaired Student *t*-test. (D) Uninjured and 5d-injured TA muscle wet weight normalized by body weight of *Ern1fl/fl* ($n = 3$) and *Ern1scKO* ($n = 4$) mice. Data information: Data were presented as mean ± SEM and analyzed by two-way ANOVA followed by Tukey's multiple comparison test. *$p \leq 0.05$; values significantly different from corresponding uninjured muscle; #$p \leq 0.05$; values significantly different from injured TA muscle of *Ern1fl/fl* mice. (E) Percentage loss in wet weight of 5d-injured TA muscle of *Ern1fl/fl* ($n = 3$) and *Ern1scKO* ($n = 4$) mice compared to contralateral uninjured muscle. Data information: Data were presented as mean ± SEM and analyzed by unpaired Student *t*-test. *$p \leq 0.05$; values significantly different from corresponding injured muscle of *Ern1fl/fl* mice. (F) Representative photomicrographs of Hematoxylin and Eosin (H&E) stained transverse sections of uninjured, 5d-, and 14d-injured TA muscle of *Ern1fl/fl* and *Ern1scKO* mice. Scale bar: 50 μm. (G, H) Quantitative analysis of average myofiber cross-sectional area (CSA) in TA muscle of *Ern1fl/fl* and *Ern1scKO* mice at (G) day 5 and (H) day 14 days post-injury. $n = 3$–5 mice in each group. Data information: Data were presented as mean ± SEM and analyzed by unpaired Student *t*-test *$p \leq 0.05$; values significantly different from injured TA muscle of *Ern1fl/fl* mice. Source data are available online for this figure.

comparable whereas the levels of IRE1α protein were significantly reduced in 5d-injured TA muscle of *Ern1scKO* mice compared with corresponding 5d-injured TA muscle of *Ern1fl/fl* mice (Fig. 3E,F). These results are consistent with our previously published report demonstrating that satellite cell-specific ablation of XBP1 (a downstream target of IRE1α) in adult mice does not affect the expression of MyoD, myogenin, or eMyHC in TA muscle at day 5 post-injury (Xiong et al, 2017). We also investigated whether the knockdown of IRE1α affects the abundance of myogenic regulatory factors (MRFs) in cultured primary myoblasts. Primary myoblasts were transfected with control or IRE1α siRNA for 24 h. The cells were then incubated in DM and collected at different time points.

Western blot analysis confirmed knockdown of IRE1α in myoblast cultures transfected with IRE1α siRNA. There was no difference in the protein levels of myogenin between control and IRE1α knockdown cultures. However, a small reduction in levels of MyHC protein was observed in IRE1α knockdown cultures compared to corresponding controls at different time points after incubation in DM (Fig. 3G).

We next sought to determine whether targeted ablation of IRE1α affects satellite cell number in the injured skeletal muscle of mice. There was no significant difference in the number of Pax7+ cells in 5d-injured TA muscle of *Ern1fl/fl* and *Ern1scKO* mice (Fig. EV2A,B). Additionally, there was no significant difference in protein levels of

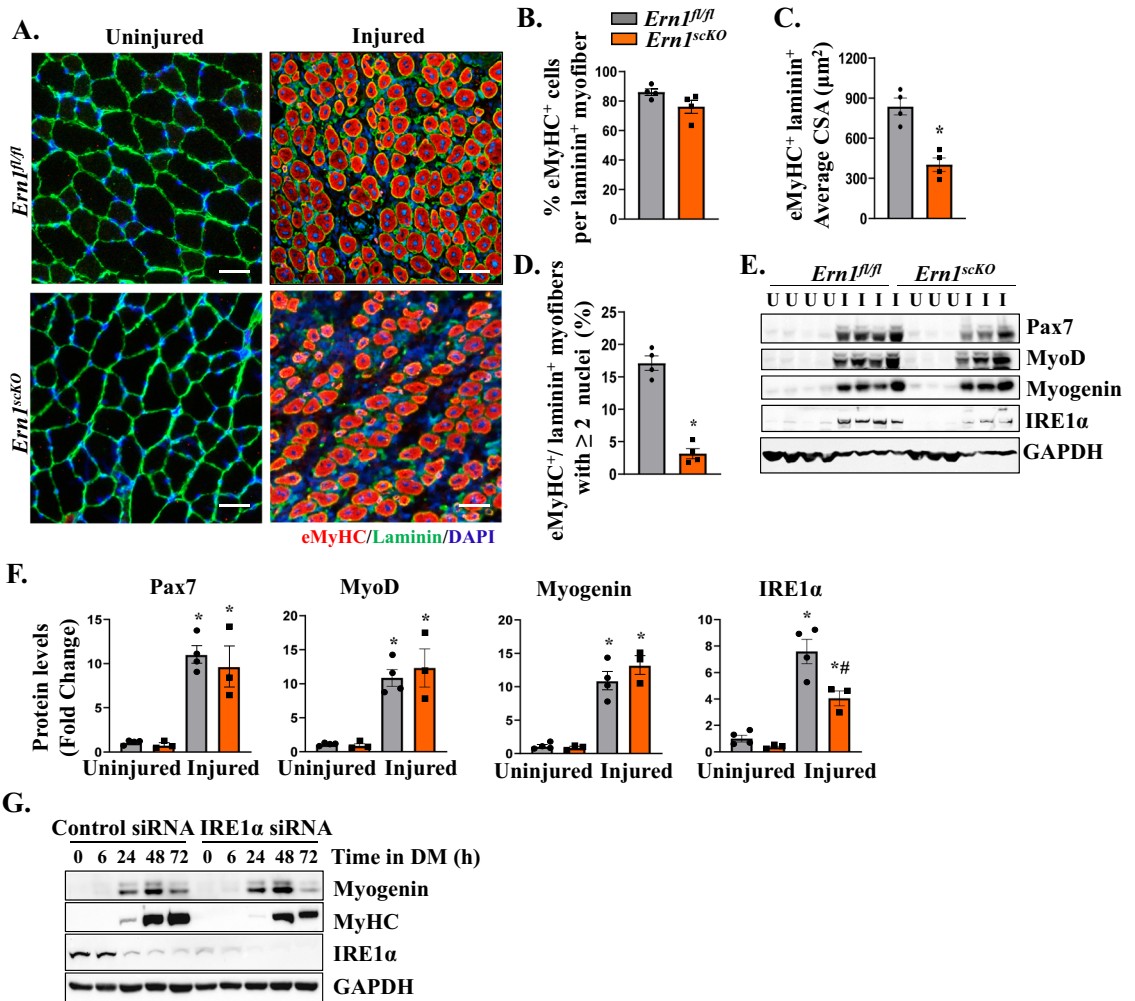

**Figure 3. Loss of IRE1α in satellite cells inhibits the formation of regenerating myofibers.**

(A) Representative photomicrographs of transverse sections of uninjured and 5d-injured TA muscle of *Ern1^{fl/fl}* and *Ern1^{scKO}* mice immunostained for an embryonic isoform of myosin heavy chain (eMyHC) and laminin protein and DAPI staining. Scale bar: 50 μm. (B–D) Quantification of (B) percentage of eMyHC⁺ cells per laminin⁺ myofibers, (C) average CSA of eMyHC⁺ laminin⁺ myofibers, and (D) percentage of eMyHC⁺ laminin⁺ myofibers containing 2 or more nuclei in 5d-injured TA muscle. (n = 3–4 mice per group). Data information: Data were presented as mean ± SEM. *p ≤ 0.05, values significantly different from corresponding muscle of *Ern1^{fl/fl}* mice analyzed by unpaired Student t-test. (E) Immunoblots showing protein levels of Pax7, MyoD, Myogenin, IRE1α, and an unrelated protein GAPDH in uninjured and 5d-injured TA muscle of *Ern1^{fl/fl}* and *Ern1^{scKO}* mice. (F) Densitometry analysis of Pax7, MyoD, Myogenin, and IRE1α immunoblots. (n = 3–4 per group). Data information: Data were presented as mean ± SEM. *p ≤ 0.05, values significantly different from corresponding uninjured muscle. #p ≤ 0.05, values significantly different from injured TA muscle of *Ern1^{fl/fl}* mice analyzed by two-way ANOVA followed by Tukey's multiple comparison test. (G) Representative immunoblots presented here show the levels of Myogenin, MyHC, IRE1α, and GAPDH protein in control or IRE1α siRNA transfected myoblasts at indicated time points after incubation in DM. U uninjured, I injured. Source data are available online for this figure.

Pax7 in uninjured or 5d-injured TA muscle of *Ern1^{fl/fl}* and *Ern1^{scKO}* mice (Fig. 3E,F). We also investigated whether siRNA-mediated knockdown of IRE1α affects the number of satellite cells in cultured myogenic cells. There was no significant difference in the proportion of Pax7⁺, MyoD⁺, or Pax7⁺/MyoD⁺ cells between control and IRE1α knockdown cultures (Fig. EV2C–F). Western blot analysis also showed that the knockdown of IRE1α does not affect the levels of Pax7 or MyoD protein in cultured myoblasts (Fig. EV2G). We have previously reported that c-Jun N-terminal kinase (JNK) phosphorylates the c-Jun transcription factor, which in turn induces the gene expression of *Pax7* in cultured myoblasts (Hindi and Kumar, 2016). Interestingly, in conditions of ER stress, IRE1α interacts with TRAF2 to phosphorylate ASK1 which in turn phosphorylates and activates

JNK1/2 (Hetz, 2012). Consistent with the levels of Pax7, there was no difference in the levels of phosphorylated or total JNK between control and IRE1α knockdown cultures (Fig. EV2G). Furthermore, the knockdown of IRE1α did not affect the proliferation of cultured myoblasts measured by EdU incorporation assay (Fig. EV2H,I). Collectively, these results suggest that satellite cell-specific deletion of IRE1α inhibits myofiber regeneration without affecting the proliferation or differentiation of satellite cells.

## Targeted deletion of IRE1α inhibits myoblast fusion

We next investigated whether targeted ablation of IRE1α affects the fusion of muscle precursor cells with injured myofibers. TA muscle

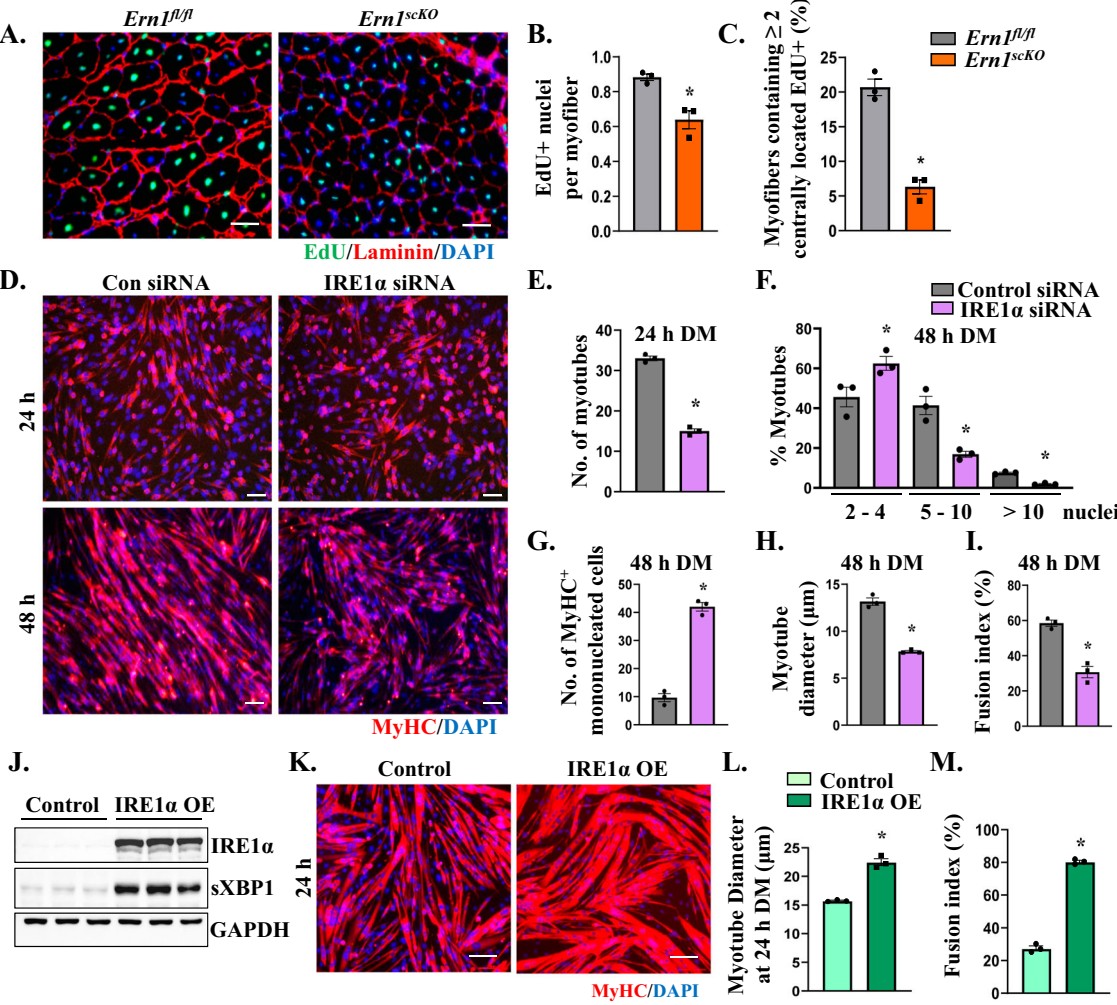

**Figure 4. IRE1α is required for myoblast fusion both in vivo and in vitro.**

(A) Representative photomicrographs of 14d-injured TA muscle sections of *Ern1^{fl/fl}* and *Ern1^{scKO}* mice after processing for detection of EdU and immunostaining for laminin protein followed by DAPI staining. Scale bar: 50 μm. (B) Quantification of EdU⁺ nuclei per myofiber in 14d-injured TA muscle sections of *Ern1^{fl/fl}* and *Ern1^{scKO}* mice. ($n = 3$ mice per group). Data information: All data were presented as mean ± SEM. *$p \leq 0.05$, values significantly different from the corresponding muscle of *Ern1^{fl/fl}* mice. (C) Quantification of the percentage of myofibers containing 2 or more centrally located EdU⁺ nuclei in 14d-injured TA muscle sections of *Ern1^{fl/fl}* and *Ern1^{scKO}* mice. ($n = 3$ mice per group). Data information: All data were presented as mean ± SEM. *$p \leq 0.05$, values significantly different from the corresponding muscle of *Ern1^{fl/fl}* mice. (D) Representative images of control siRNA or IRE1α siRNA transfected myoblast cultures at 24 or 48 h after the addition of DM, followed by immunostaining for MyHC protein and DAPI staining. Scale bar: 50 μm. (E–I) Quantitative analysis of (E) number of myotubes at 24 h, (F) percentage of myotubes containing 2–4, 5–10, and more than 10 nuclei at 48 h, (G) number of MyHC⁺ mononucleated cells at 48 h, (H) average myotube diameter, and (I) fusion index in control and IRE1α siRNA transfected cultures at 48 h after incubation in DM. $n = 3$ (biological replicates) per group. Data information: All data were presented as mean ± SEM. *$p \leq 0.05$, values significantly different from control cultures analyzed by unpaired Student *t*-test. (J) Representative immunoblots presented here show levels of IRE1α and sXBP1, and an unrelated protein GAPDH in control and IRE1α overexpressing (OE) cultures. (K) Representative images of control and IRE1α OE myoblast cultures incubated in DM for 24 h followed by immunostaining for MyHC protein and DAPI staining. Scale bar: 50 μm. (L, M) Quantification of (L) average diameter of MyHC⁺ cells and (M) fusion index. $n = 3$ (biological replicates) per group. Data information: All data were presented as mean ± SEM. *$p \leq 0.05$, values significantly different from control cultures analyzed by unpaired Student *t*-test. Source data are available online for this figure.

of *Ern1^{fl/fl}* and *Ern1^{scKO}* mice was injured using intramuscular injection of 1.2% BaCl₂ solution. After 72 h, the mice were given a single intraperitoneal injection of EdU. Finally, TA muscle was collected on day 14 post-injury, followed by the analysis of EdU⁺ nuclei. Results showed that the number of EdU⁺ nuclei per myofiber and the proportion of myofibers containing two or more centrally located EdU⁺ nuclei were significantly reduced in injured TA muscle of *Ern1^{scKO}* mice compared to corresponding *Ern1^{fl/fl}* mice (Fig. 4A–C) suggesting fusion defects in the regenerating

muscle of *Ern1^{scKO}* mice. Next, we investigated whether IRE1α affects the fusion of cultured myoblasts. Primary myoblasts prepared from WT mice were transfected with control or IRE1α siRNA. The cells were then incubated in DM for 24 or 48 h, followed by immunostaining for myosin heavy chain (MyHC) protein and DAPI staining. Results showed that the knockdown of IRE1α significantly reduced the number of myotubes (containing two or more nuclei) at 24 h after the addition of DM (Fig. 4D,E). At 48 h of addition of DM, we found that the proportion of myotubes

containing 2–4 nuclei was significantly increased, whereas the proportion of myotubes containing 5–10 nuclei or more than 10 nuclei was significantly reduced in IRE1α knockdown cultures compared to control cultures (Fig. 4D,F). Indeed, the number of MyHC[+] mononucleated cells was also significantly increased in IRE1α silenced cultures compared to corresponding control cultures at 48 h of incubation in DM (Fig. 4G). In addition, average myotube diameter and fusion index was significantly reduced in IRE1α knockdown cultures compared to control cultures (Fig. 4H,I). Western blot analysis showed that while the levels of MyoD and MyHC were comparable, the levels of IRE1α protein were drastically reduced in myoblast cultures transfected with IRE1α siRNA compared to control cultures at 24 h of addition of DM (Fig. EV3A).

We next investigated the effects of overexpression of IRE1α in cultured myoblasts. Previous studies have demonstrated that overexpression of IRE1α is sufficient for its activation in cultured mammalian cells in the absence of ER stress (Acosta-Alvear et al, 2018; Hollien et al, 2009). Primary myoblasts were transduced with retrovirus expressing a cDNA encoding IRE1α or enhanced green fluorescence protein (EGFP, as control). By performing a Western blot, we first confirmed that the levels of IRE1α protein are increased in cultures transduced with IRE1α cDNA encoding retrovirus compared to those transduced with EGFP retrovirus. Intriguingly, there was also a considerable increase in the levels of sXBP1 protein in IRE1α overexpressing (OE) cultures (Fig. 4J). Next, control and IRE1α OE cultures were incubated in DM for 24 h followed by immunostaining for MyHC protein. Results showed that overexpression of IRE1α improved the formation of myotubes with an increased diameter (Fig. 4K,L). Furthermore, the fusion index of IRE1α OE cultures was significantly increased compared to control cultures incubated in DM (Fig. 4M) suggesting that forced activation of IRE1α improves myoblast fusion.

Recently, a pharmacological compound named IXA4 was identified as an efficient and selective activator of the IRE1α/sXBP1 signaling axis (Grandjean et al, 2020). We investigated whether pharmacological activation of IRE1α using IXA4 compound increases the fusion of cultured myoblasts. We first determined the concentration of IXA4 that is required for augmenting the levels of sXBP1 protein in cultured myoblasts (Fig. EV3B). Finally, primary myoblasts were treated with IXA4 and incubated in DM for 48 h followed by immunostaining for MyHC protein. Intriguingly, the average myotube diameter as well as fusion index were significantly increased in cultures treated with IXA4 compared to control cultures treated with vehicle alone (Fig. EV3C–E). In another experiment, we also investigated whether IRE1α-mediated signaling is required in one or both fusion partners during myogenesis. For this experiment, we generated lentiviral particles expressing either a scrambled shRNA or IRE1α shRNA along with mCherry protein. In addition, we used lentiviral particles expressing GFP protein alone. Primary myoblasts were stably transduced with lentiviral particles expressing GFP protein alone (control myoblasts) or scrambled or IRE1α shRNA along with mCherry. Finally, an equal number of control and scrambled shRNA or IRE1α shRNA-expressing myoblasts were mixed and incubated in DM for 48 h. Results showed that the knockdown of IRE1α in myoblasts considerably reduced their fusion with control myoblasts (Fig. EV3F), suggesting that IRE1α is required in both fusion partners for efficient myotube formation.

Non-canonical NF-κB and canonical Wnt signaling play important roles in myoblast fusion during myogenesis (Brack et al, 2008; Hayden and Ghosh, 2004; Hindi et al, 2017; Razani et al, 2011). To understand the signaling mechanisms by which IRE1α promotes myoblast fusion, we investigated the effects of knockdown of IRE1α on the markers of canonical and non-canonical NF-κB and Wnt signaling at 48 h of incubation in DM. However, no difference was observed in the levels of MyHC (a marker of muscle differentiation), p-p65 and p65 (markers of canonical NF-κB), p100 and p52 (markers of non-canonical NF-κB) and p-GSK-3β (a marker of Wnt signaling) protein in control and IRE1α knockdown cultures (Fig. EV3G). Furthermore, there was no significant difference in the mRNA levels of Wnt ligands (Wnt4, Wnt5a, Wnt3a, and Wnt11), Wnt receptors (Fzd2, Fzd4, and Fzd6), and target gene Axin2 in control and IRE1α knockdown cultures (Figure EV3H). These results suggest that IRE1α promotes myoblast fusion without affecting the activation of NF-κB and Wnt signaling during myogenesis.

## Transcriptome analysis corroborates the role of IRE1α in myoblast fusion

By performing RNA sequencing (RNA-Seq), we next studied global gene expression changes in control and IRE1α knockdown myoblast cultures at 24 h after the addition of DM. Differentially Expressed Genes (DEGs) were classified based on at least 1.5 times fold change from the basal levels of control cells coupled with the criteria for significance of $p \leq 0.05$. Analysis of DEGs revealed that 521 genes were downregulated whereas 555 genes were upregulated in IRE1α knockdown cultures compared with controls (Fig. 5A). Pathway analysis using Metascape Gene Annotation and Analysis Tool showed the involvement of downregulated gene sets in the processes, such as regulation of protein exit from ER, response to topologically incorrect proteins, and vesicle localization. By contrast, upregulated gene sets were associated with the regulation of nuclease activity, muscle system process, negative regulation of response to external stimulus, and regulation of vasculature development and angiogenesis (Fig. 5B). Downregulation of protein-folding related genes comply with the established role of IRE1α in the resolution of ER stress (Gardner and Walter, 2011). ER stress-mediated activation of IRE1α exerts its downstream effects through three identified mechanisms: JNK-mediated signaling, XBP1 mRNA splicing, and RIDD pathway (Hollien et al, 2009; Maurel et al, 2014; Urano et al, 2000; Wang and Kaufman, 2014). Our preceding results showed that the knockdown of IRE1α does not affect the activation of JNK in cultured myoblasts (Fig. EV2G). Recent studies have identified a few mRNA in skeletal muscle and other cell types subjected to the RIDD pathway (Quwaider et al, 2022). However, our RNA-Seq analysis showed no significant alterations in selected putative substrates of the RIDD pathway between control and IRE1α knockdown cultures at 24 h after incubation in DM (Fig. EV4A). We also measured the mRNA levels of RIDD substrates (e.g., Bloc1s1, Erdj4, Pdgfr, Scara3, and Sparc) between control and IRE1α knockdown cultures at 48 h after incubation in DM. While the levels of Pdgfr, Scara3, and sXbp1 were reduced, there was no significant reduction in the mRNA levels of Bloc1s1, Erdj4, and Sparc) between control and IRE1α knockdown cultures (Fig. EV4B), suggesting that IRE1α does not regulate the RIDD pathway during myogenic differentiation.

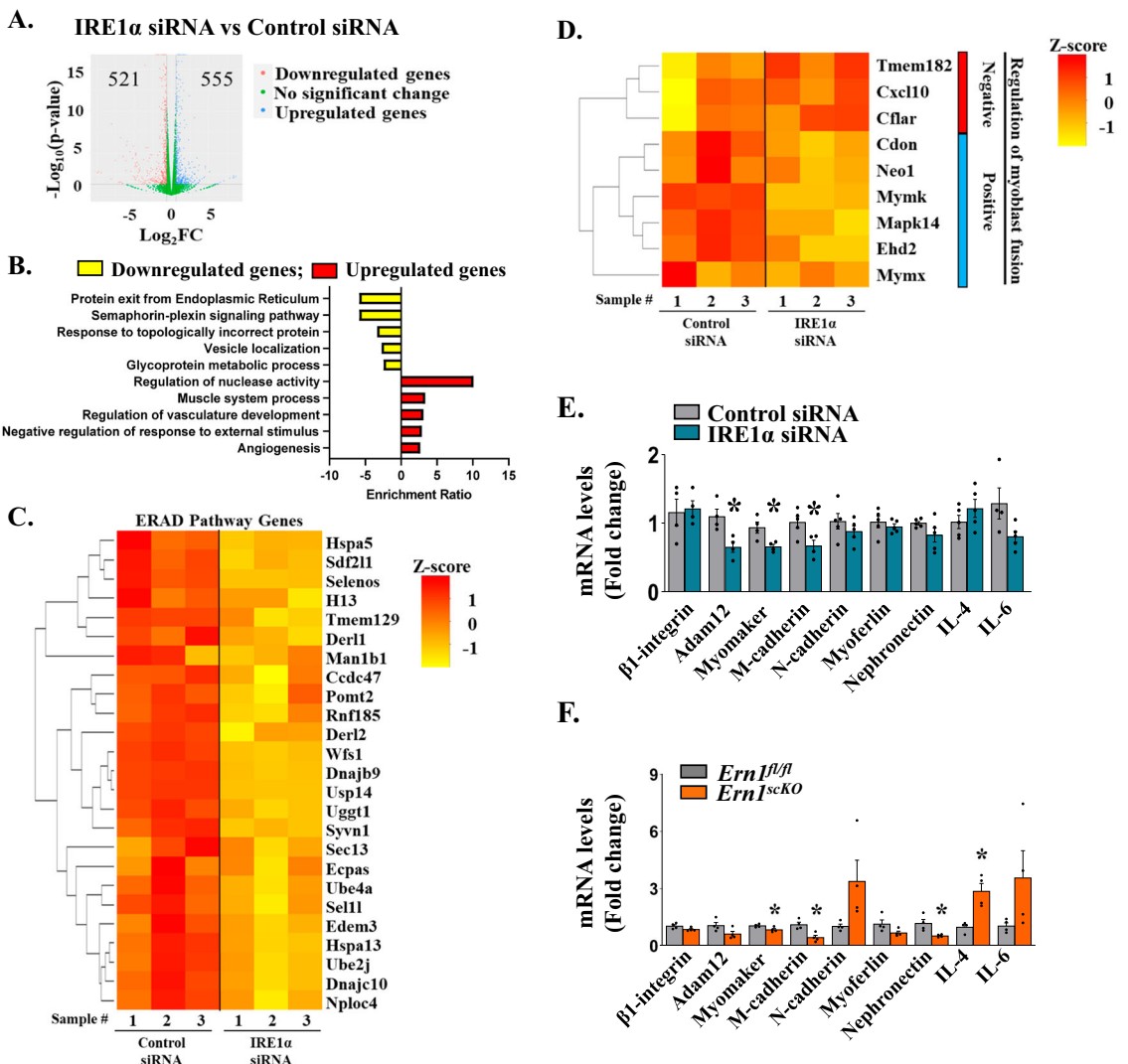

**Figure 5. Global transcriptomic changes in IRE1α deficient myoblast cultures.**

(A) Volcano plot showing the distribution of upregulated, downregulated, and unaltered genes in control siRNA or IRE1α siRNA transfected mouse primary myoblast cultures incubated in DM for 24 h. $n = 3$ (biological replicates) per group. Data information: Differential expression of genes was analyzed by using threshold values of Log2FC≥|0.25| and $p$ value <0.05; and annotated by red (downregulated), green (no significant change), and blue (upregulated) colors. (B) Enriched gene ontology (GO) Biological processes and Reactome pathways associated with the upregulated and downregulated gene sets analyzed using the Metascape Gene Annotation and Analysis tool. (C) Heatmap representing z-scores (based on transcript per million reads) of genes associated with the ERAD pathway compared between control- and IRE1α-siRNA groups. (D) Heatmap representing selected genes involved in the regulation of myoblast fusion. (E) Relative mRNA levels of profusion molecules in control and IRE1α knockdown cultures at 24 h of incubation in DM. $n = 4$ (biological replicates) per group. Data information: All data were presented as mean ± SEM. *$p \le 0.05$, values significantly different from control cultures analyzed by unpaired Student $t$-test. (F) Relative mRNA levels of various profusion molecules in 5d-injured TA muscle of $Ern1^{fl/fl}$ and $Ern1^{scKO}$ mice. $n = 4$ mice per group. Data information: All data were presented as mean ± SEM. *$p \le 0.05$, values significantly different from injured muscle of $Ern1^{fl/fl}$ mice analyzed by unpaired Student $t$-test. Source data are available online for this figure.

Therefore, we focused our further investigation to determine whether the knockdown of IRE1α impairs myoblast fusion through sXBP1-mediated transcriptional regulation. It is known that sXBP1 regulates the gene expression of several molecules which are involved in multiple pathways, including but not limited to protein-folding, protein entry into the ER, and ER-associated degradation (ERAD) pathway (Park et al, 2021; Wu et al, 2015). Heatmaps generated using z-scores based on transcript per million (TPM) values showed diminished transcript levels of multiple molecules associated with the ERAD pathway (Fig. 5C). We next investigated

whether IRE1α regulates the gene expression of ERAD molecules in myoblasts through XBP1 transcription factor. For this experiment, primary myoblasts were transfected with control or XBP1 siRNA followed by incubation in DM for 24 h and performing RNA-Seq analysis. Remarkably, the mRNA levels of the majority of ERAD-related molecules downregulated in IRE1α knockdown cultures were also reduced in XBP1 knockdown myoblast cultures (Fig. EV4C).

We next investigated whether the knockdown of IRE1α or XBP1 affects the gene expression of molecules that mediate myoblast

fusion. We found that siRNA-mediated knockdown of IRE1α or XBP1 resulted in deregulation of both positive and negative regulators of myoblast fusion (Figs. 5D and EV4D). Interestingly, mRNA levels of myomaker (*Mymk*), a critical regulator of myoblast fusion, were significantly reduced in IRE1α or XBP1 knockdown cultures compared to corresponding controls (Figs. 5D and EV4D). In a separate experiment, we also measured relative mRNA levels of a few other profusion molecules in control and IRE1α knockdown myoblast cultures at 24 h of incubation in DM. Results showed that mRNA levels of Adam12, myomaker, and M-cadherin were significantly reduced in IRE1α knockdown cultures compared to controls (Fig. 5E). In addition, our qRT-PCR analysis showed that mRNA levels of some of these profusion molecules (i.e., myomaker, M-Cadherin, and nephronectin) were significantly reduced in 5d-injured TA muscle of *Ern1scKO* mice compared to corresponding *Ern1fl/fl* mice (Fig. 5F). We also investigated the effect of overexpression of IRE1α on the transcript levels of various profusion molecules. Interestingly, the mRNA levels of myomaker, myomerger, M-cadherin, Caveolin-3, β1D-integrin, and nephronectin were significantly increased in IRE1α overexpressing myoblast cultures compared to controls at 24 h of addition of DM (Fig. EV5A). Collectively, these results suggest that IRE1α-mediated signaling promotes myoblast fusion by augmenting the gene expression of various profusion molecules, including myomaker.

## IRE1α augments levels of sXBP1 and myomaker during myogenesis

We next investigated whether IRE1α promotes myoblast fusion by augmenting the levels of transcriptionally active sXBP1 protein. Primary myoblasts prepared from WT mice were transfected with control or XBP1 siRNA followed by incubation in DM for 24 or 48 h. The myotube formation was evaluated by immunostaining for MyHC protein (Fig. 6A). Interestingly, myotube formation was significantly reduced in myoblast cultures transfected with XBP1 siRNA at 24 h after incubation in DM (Fig. 6A,B). At 48 h, the number of myotubes with 2–4 nuclei was significantly increased whereas the number of myotubes containing 5–10 or more than 10 nuclei was significantly reduced in cultures transfected with XBP1 siRNA compared to control cultures (Fig. 6C). Moreover, the number of MyHC⁺ mononucleated cells was significantly increased in XBP1 siRNA transfected cultures (Fig. 6D). In addition, there was a significant reduction in the average myotube diameter and fusion index in XBP1 knockdown cultures compared to control cultures at 48 h of addition of DM (Fig. 6A,E,F).

Western blot analysis revealed that the protein levels of MyHC were comparable between control and XBP1 knockdown cultures (Fig. EV5B), suggesting that siRNA-mediated knockdown of XBP1 inhibits myotube formation without affecting myogenic differentiation. While silencing of IRE1α reduces the levels of sXBP1, our analysis showed that knockdown of XBP1 does not affect the levels of p-IRE1α or total IRE1α protein in cultured myoblasts after incubation in DM (Fig. EV5C). In another experiment, we also investigated the effects of pharmacological inhibition of kinase or endonuclease activity of IRE1α on myoblast fusion. APY29 is a kinase inhibitor that inhibits IRE1α autophosphorylation and enhances its RNase function. B-I09 is an IRE1α RNase inhibitor

that is highly effective in inhibiting splicing of XBP1 mRNA in mammalian cells. Similarly, 4μ8C is a potent and selective IRE1α RNase inhibitor (Hetz et al, 2019). We first confirmed that treatment with APY29 increases whereas B-I09 and 4μ8C reduces the levels of sXBP1 in cultured primary myoblasts (Fig. EV5D). Next, the primary myoblasts were incubated in DM with vehicle alone or with APY29, B-I09, or 4μ8C, and the myotube formation was evaluated 24 h later. Interestingly, treatment with APY29 significantly increased the formation of myotubes. By contrast, a significant reduction in myotube formation and fusion index was observed in the cultures treated with B-I09 or 4μ8C (Fig. EV5E–G).

Transmembrane protein myomaker plays an essential role in myoblast fusion (Millay et al, 2013; Millay et al, 2014). Our RNA-Seq analysis revealed that mRNA levels of myomaker are reduced following the knockdown of IRE1α or XBP1 in cultured myoblasts. We next investigated whether IRE1α increases the expression of myomaker through augmenting levels of sXBP1 protein. Results showed that siRNA-mediated knockdown of IRE1α reduced the levels of both sXBP1 and myomaker protein in cultured myoblasts incubated in DM for 24 h (Fig. 6G). Similarly, siRNA-mediated knockdown of XBP1 also reduced the amount of myomaker protein in cultured myoblast at 24 h of incubation in DM (Fig. 6H). We next investigated the effect of overexpression of sXBP1 on myotube formation. Interestingly, retroviral-mediated overexpression of sXBP1 improved myotube formation, average myotube diameter, and fusion index at 24 h of incubation in DM (Fig. 6I–L). In another experiment, we investigated how overexpression of IRE1α affects the levels of sXBP1 and myomaker protein in cultured myoblasts. Results showed that the levels of both myomaker and sXBP1 protein were increased in IRE1α overexpressing cultured myoblasts (Fig. 6M). Taken together, these results suggest that the activation of IRE1α induces myoblast fusion by enhancing the levels of sXBP1 protein, which induces the gene expression of myomaker.

## Transcription factor sXBP1 binds to the promoter region of the *Mymk* gene

The sXBP1 protein binds to the promoter regions of multiple genes to regulate their transcription. We first determined whether the promoter region of the *Mymk* gene contains a consensus core DNA sequence for the sXBP1 transcription factor. Specifically, the UCSC genome browser was used to locate and identify conserved binding domains of sXBP1 in the regulatory region of the *Mymk* gene in the Mouse GRCm39/mm39 genome database. The ReMap ChIP-Seq function is an integrative atlas of multiple ChIP-Seq analyses deposited on the Gene Expression Omnibus database (Cheneby et al, 2018; Cheneby et al, 2020; Griffon et al, 2015; Hammal et al, 2022). Interestingly, we observed a predicted sXBP1 binding site in the proximity of the *Mymk* gene, which is also described by Liu et al (Liu et al, 2019) for ER transporter protein genes, such as *Sec23b*, *Sec61a1*, and *Sec61b*. To identify these genome sequences containing the core-binding domain of sXBP1, the GenBank sequence of the two predicted isoforms of *Mymk* gene, flanking the region 26951648-26962173 on chromosome 2 (GRCm39), were manually searched for the "ACGT" core-binding sXBP1 domain (Fig. 7A). We identified two sites, named Binding-Site1 and Binding-Site2 localized after exon1 of *Mymk* predicted isoform 1 or ~4000 bp for

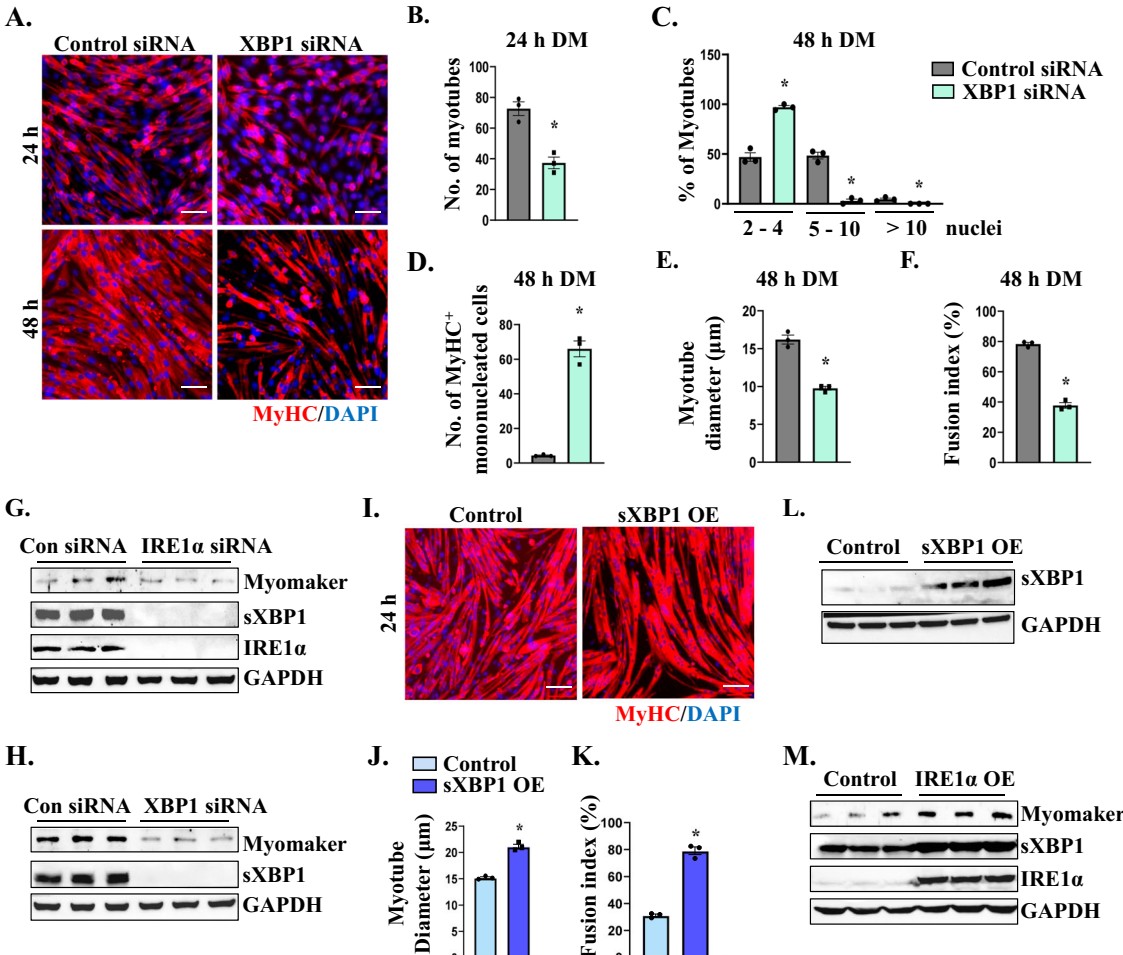

**Figure 6. IRE1α regulates myoblast fusion through XBP1.**

(A) Representative photomicrographs of control and XBP1 siRNA transfected cultures incubated in DM for 24 or 48 h followed by immunostaining for MyHC protein and staining with DAPI. Scale bar: 50 μm. (B–F) Quantitative analysis of (B) number of myotubes at 24 h, (C) percentage of myotubes containing 2–4, 5–10, or more than 10 nuclei at 48 h, (D) number of MyHC⁺ mononucleated cells at 48 h, (E) average myotube diameter at 48 h, and (F) fusion index at 48 h after addition of DM in control and XBP1 knockdown cultures. $n = 3$ (biological replicates) per group. Data information: All data were presented as mean ± SEM. *$p ≤ 0.05$, values significantly different from control cultures analyzed by unpaired Student $t$-test. (G) Immunoblots presented here show protein levels of myomaker, sXBP1, IRE1α, and unrelated protein GAPDH in cultures transfected with control or IRE1α siRNA and incubated in DM for 24 h. (H) Immunoblots presented here show the levels of myomaker, sXBP1, and GAPDH in cultures transfected with control or XBP1 siRNA at 24 h of incubation in DM. (I) Representative photomicrographs of control and sXBP1-overexpressing (OE) cultures incubated in DM for 24 h and immunostained for MyHC protein. Nuclei were stained with DAPI. Scale bar: 50 μm. (J, K) Quantification of (J) average myotube diameter and (K) fusion index in control and sXBP1 OE cultures. $n = 3$ (biological replicates) per group. Data information: All data were presented as mean ± SEM. *$p ≤ 0.05$, values significantly different from control cultures analyzed by unpaired Student $t$-test. (L) Immunoblots presented here show protein levels of sXBP1 and unrelated protein GAPDH in control and sXBP1 OE cultures. (M) Immunoblots presented here show the levels of myomaker, sXBP1, IRE1α, and unrelated protein GAPDH in control and IRE1α OE cultures incubated in DM for 24 h. Source data are available online for this figure.

Site1 and ~2100 bp for Site2 upstream of *Mymk* predicted isoform 2 (Fig. 7B). To determine whether sXBP1 binds to these sequences during myogenesis, we performed chromatin immunoprecipitation (ChIP) assay followed by semi-quantitative PCR and quantitative real time-PCR assays using primer sets designed to amplify 100–150 bp genome sequence flanking predicted DNA binding Site1 or Site2 in *Mymk* gene. Results showed increased enrichment of sXBP1 to both predicted sites in the *Mymk* gene (Fig. 7C,D). The ChIP experiment was confirmed using standard PCRs for positive control (Rpl30 for Histone H3 ChIP antibody) and known sXBP1 target sequence in promoter regions of Hspa5 and Dnajb9 genes (Fig. 7C). We next studied the effect of overexpression of sXBP1 on

the transactivation of myomaker promoter. We generated a luciferase reporter using ~2100 bp sequence upstream of the transcription start site of Mymk transcript variant 2 that consisted of binding site 2. Consistent with ChIP results, we found that the overexpression of sXBP1 significantly increased the reporter gene activity in mouse primary myoblasts incubated in DM for 24 h (Fig. 7E).

To understand whether IRE1α-XBP1 signaling induces myoblast fusion through augmenting the gene expression of myomaker, we investigated whether forced expression of myomaker can rescue fusion defects in IRE1α knockdown cultures. Interestingly, overexpression of myomaker significantly improved myotube

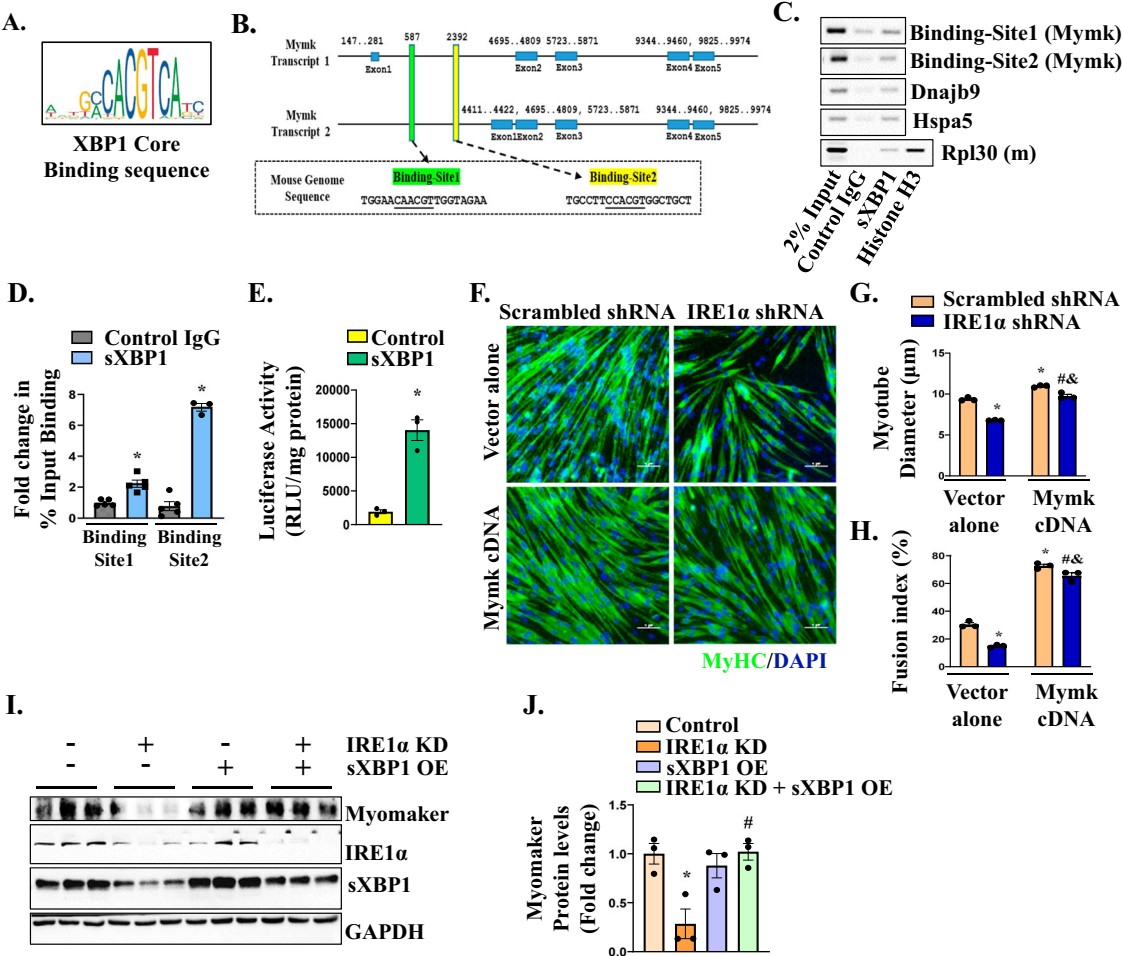

**Figure 7. sXBP1 regulates the gene expression of myomaker during myoblast fusion.**

(A) Symbolic representation of the core-binding sequence of sXBP1 transcription factor (downloaded from JASPER website). (B) Schematic representation of mouse *Mymk* genomic sequence that includes sXBP1 binding Site1 and Site2. (C) Agarose gel images of PCR show levels of enrichment of *sXBP1* at indicated sXBP1 binding sites in *Mymk* gene and positive control Rpl30 against Histone H3 antibody and known sXBP1 targets Dnajb9 and Hspa5 in cultured primary myoblasts. (D) ChIP assay followed by quantitative PCR (qPCR) analysis depicting the percentage of input enrichment of sXBP1 at putative Site 1 and Site 2 in *Mymk* gene in cultured myoblasts. $n = 3$–5 (biological replicates) per group. Data information: Data were presented as mean ± SEM. *$p \leq 0.05$; values significantly different from control IgG analyzed by unpaired Student $t$-test. (E) Quantification of promoter-reporter (i.e., luciferase) activity in control and sXBP1-overexpressing (OE) myoblast cultures at 24 h of addition of DM. $n = 3$ (biological replicates) per group. Data information: Data were presented as mean ± SEM. *$p \leq 0.05$, values significantly different from control cultures analyzed by unpaired Student $t$-test. (F) Representative photomicrographs of the myoblasts cultures stably expressing scrambled shRNA or IRE1α shRNA and transfected with empty vector or plasmid containing myomaker cDNA and incubated in DM for 48 h followed immunostaining for MyHC protein and DAPI staining. Scale bar: 50 μm. (G, H) Quantification of (G) average myotube diameter and (H) fusion index in control and myomaker overexpressing cultures. $n = 3$ (biological replicates) per group. Data information: Data were presented as mean ± SEM. *$p \leq 0.05$, values significantly different from corresponding control cultures, #$p \leq 0.05$, values significantly different from cultures transfected with IRE1α shRNA alone, and &$p \leq 0.05$, values significantly different from cultures transfected with scrambled shRNA and myomaker cDNA analyzed by two-way ANOVA followed by Tukey's multiple comparison test. (I) Immunoblots presented here show levels of myomaker, IRE1α, sXBP1, and GAPDH protein in IRE1α knockdown cultures overexpressing sXBP1. (J) Densitometry analysis of levels of myomaker protein in IRE1α knockdown cultures overexpressing sXBP1. $n = 3$ (biological replicates) per group. Data information: Data were presented as mean ± SEM. *$p \leq 0.05$, values significantly different from control cultures and #$p \leq 0.05$, values significantly different from IRE1α KD cultures analyzed by two-way ANOVA followed by Tukey's multiple comparison test. KD knockdown, OE overexpression. Source data are available online for this figure.

diameter and fusion index in IRE1α knockdown cultures (Figs. 7F–H and EV5H,I). We next investigated whether over-expression sXBP1 increases the levels of myomaker in IRE1α knockdown cultures. Interestingly, overexpression of sXBP1 significantly increased the levels of myomaker protein in IRE1α knockdown cultures (Fig. 7I,J). Collectively, these results suggest that the IRE1α-XBP1 signaling promotes myoblast fusion by augmenting the gene expression of myomaker.

## IRE1α regulates myoblast fusion during overload-induced muscle growth

In addition to muscle regeneration, myoblast fusion is also essential for myofiber hypertrophy and skeletal muscle growth in response to functional overload (Goh and Millay, 2017; Hindi et al, 2017). To confirm the physiological significance of IRE1α/XBP1 signaling in myoblast fusion in adult mice, we investigated whether IRE1α/

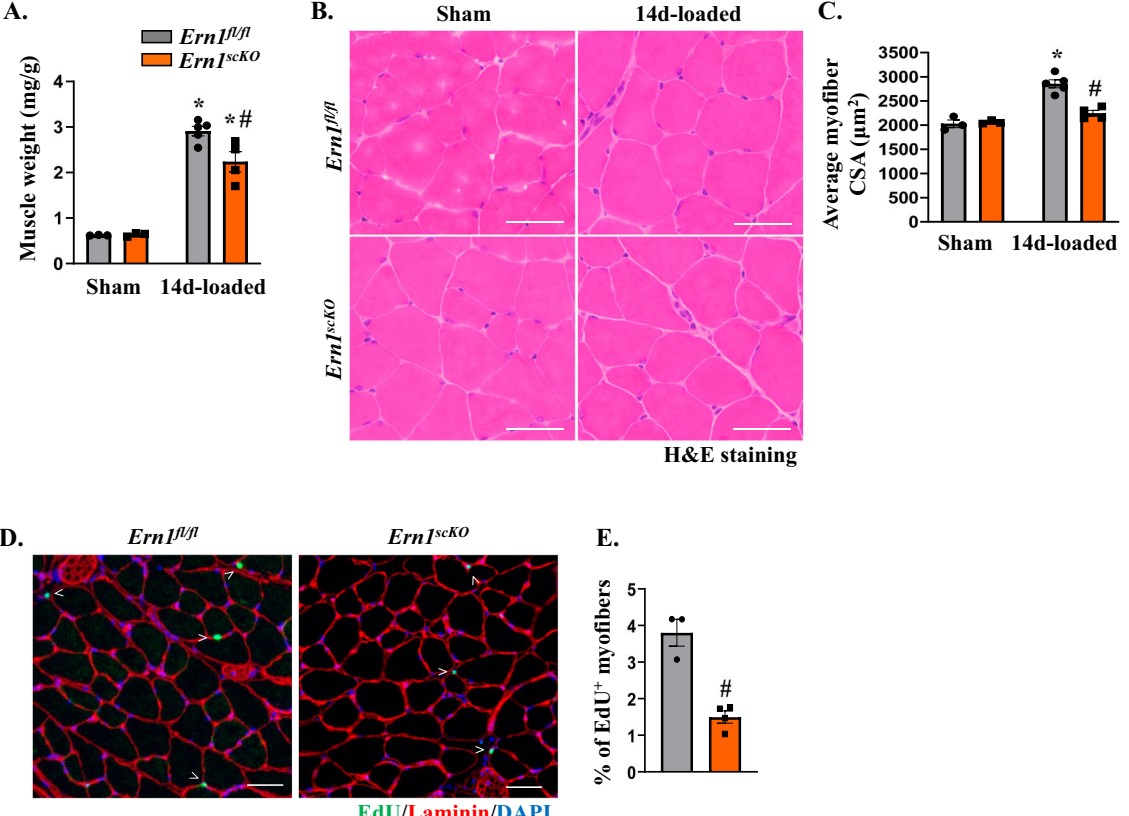

**Figure 8. IRE1α promotes myoblast fusion during overload-induced muscle hypertrophy.**

(A) Quantification of the wet weight of plantaris muscle of *Ern1$^{fl/fl}$* and *Ern1$^{scKO}$* mice subjected to sham or synergistic ablation (SA) surgery normalized to body weight. $n = 3$–5 mice per group. Data information: Data were presented as mean ± SEM and analyzed by two-way ANOVA followed by Tukey's multiple comparison test. *$p \leq 0.05$; values significantly different from sham operated of *Ern1$^{fl/fl}$* mice. #$p \leq 0.05$; values significantly different from 14d-overloaded plantaris muscle of *Ern1$^{fl/fl}$* mice. (B) Representative photomicrographs of H&E-stained transverse sections of sham or 14d-loaded plantaris muscle of *Ern1$^{fl/fl}$* and *Ern1$^{scKO}$* mice. Scale bar: 50 μm. (C) Quantitative analysis of average myofibers CSA of sham and 14d-loaded plantaris muscle of *Ern1$^{fl/fl}$* and *Ern1$^{scKO}$* mice. $n = 3$–5 mice per group. Data information: Data were presented as mean ± SEM and analyzed by two-way ANOVA followed by Tukey's multiple comparison test. *$p \leq 0.05$; values significantly different from sham operated of *Ern1$^{fl/fl}$* mice. #$p \leq 0.05$; values significantly different from 14d-overloaded plantaris muscle of *Ern1$^{fl/fl}$* mice. (D) Representative images of 14d-loaded plantaris muscle of *Ern1$^{fl/fl}$* and *Ern1$^{scKO}$* mice after staining for EdU and immunostaining for laminin protein followed by DAPI staining. Scale bar: 50 μm. (E) Quantification of the proportion of EdU$^+$ myofibers in 14d-loaded plantaris muscle of *Ern1$^{fl/fl}$* and *Ern1$^{scKO}$* mice. $n = 3$–4 mice per group. Data information: Data were presented as mean ± SEM and analyzed by unpaired Student $t$-test #$p \leq 0.05$; values significantly different from 14d-overloaded plantaris muscle of *Ern1$^{fl/fl}$* mice. Source data are available online for this figure.

XBP1 signaling also promotes myoblast fusion during overload-induced muscle hypertrophy. We used a bilateral synergistic ablation (SA) model that has been consistently used to induce hypertrophy of the plantaris muscle of adult mice (Hindi et al, 2018; Hindi et al, 2017). After 2 days of sham or SA surgery, the mice were given an intraperitoneal injection of EdU and plantaris muscle was collected on day 14 of SA surgery. There was a` significant increase in the wet weight of plantaris muscle in both *Ern1$^{fl/fl}$* and *Ern1$^{scKO}$* mice at 14d of performing SA surgery. However, gain in wet weight of plantaris muscle in response to functional overload normalized by body weight was significantly reduced in *Ern1$^{scKO}$* mice compared with *Ern1$^{fl/fl}$* mice (Fig. 8A). Next, we generated transverse sections of plantaris muscle and performed H&E staining followed by analysis of myofiber CSA. Results showed that the average myofiber CSA of 14d-overloaded plantaris muscle was significantly reduced in *Ern1$^{scKO}$* mice compared to *Ern1$^{fl/fl}$* mice (Fig. 8B,C). To determine whether the reduction in myofiber CSA in plantaris muscle of *Ern1$^{scKO}$* mice in

response to functional overload is attributed to a deficit in myoblast fusion, we analysed plantaris muscle sections for EdU incorporation in myofibers (Fig. 8D). Interestingly, the percentage of myofibers containing EdU$^+$ nuclei in plantaris muscle was significantly reduced in *Ern1$^{scKO}$* mice compared to *Ern1$^{fl/fl}$* mice (Fig. 8E). These results further suggest that IRE1α promotes skeletal muscle growth in adult mice through augmenting myoblast fusion.

## Discussion

Skeletal muscle regeneration is regulated by a complex network of signaling pathways. In addition to pro-myogenic signaling, accumulating evidence suggests that skeletal muscle regeneration involves the activation of UPR pathways (Afroze and Kumar, 2019; Bohnert et al, 2018; Roy et al, 2021). We recently reported that the IRE1α/XBP1 arm of the UPR is highly activated in skeletal muscle in response to injury and that myofiber-specific deletion of IRE1α

or XBP1 attenuates muscle regeneration potentially through inhibiting the proliferation of satellite cells in a cell non-autonomous manner (Roy et al, 2021). However, the cell-autonomous role and mechanisms of action of IRE1α in muscle progenitor cells remained unknown. In the present study, our results demonstrate that IRE1α promotes myoblast fusion by enhancing the levels of sXBP1 protein and augmenting the gene expression of myomaker, a critical regulator of myoblast fusion.

Recent studies from our group and others have highlighted the role of UPR pathways in muscle progenitor cell function and myogenic differentiation. It has been found that the PERK/eIF2α arm of the UPR is essential for the survival and self-renewal of satellite cells in the skeletal muscle of adult mice (Afroze and Kumar, 2019; Xiong et al, 2017; Zismanov et al, 2016). Loss of PERK drastically reduces the pool of satellite cells in injured myofibers resulting in a significant decrease in various parameters of muscle regeneration, including the gene expression of myogenic regulatory factors (MRFs), MyoD and Myogenin and the levels of eMyHC (Xiong et al, 2017). In contrast, satellite cell-specific ablation of XBP1 in mice did not affect the expression of MRFs or eMyHC in regenerating skeletal muscle suggesting that XBP1 does not affect the myogenic function of satellite cells in adults. Since the markers of muscle differentiation were not affected, we did not perform an extensive analysis of muscle regeneration phenotype in satellite cell-specific XBP1-knockout mice. However, our limited analysis showed a trend toward reduction in average myofiber CSA and the number of myofibers containing two or more centrally located nuclei in the TA muscle of satellite cell-specific XBP1-knockout mice at day 5 post-injury (Xiong et al, 2017).

Proliferating satellite cells eventually differentiate into myoblasts/myocytes, which fuse with each other or damaged myofibers to accomplish muscle repair. Our results using loss-of-function and gain-of-function approaches demonstrate that IRE1α-mediated signaling promotes myoblast fusion (Figs. 4 and EV3). Myogenic differentiation is tightly controlled by the sequential expression of various MRFs. Consistent with results in satellite cell-specific XBP1-knockout mice (Xiong et al, 2017), there was no significant difference in the protein levels of MyoD and myogenin in regenerating skeletal muscle of *Ern1^{fl/fl}* and *Ern1^{scKO}* mice. Similarly, the knockdown of IRE1α in cultured myoblasts did not affect the levels of MRFs or MyHC protein at different time points after induction of differentiation (Figs. 3 and EV3) suggesting that the deficiency of IRE1α or XBP1 in satellite cells specifically inhibits myoblast fusion without having any effect on the myogenic differentiation program.

Myoblast fusion is also critical for overload-induced myofiber hypertrophy in adults (Goh and Millay, 2017; Hindi and Millay, 2022; Hindi et al, 2013). Our results demonstrate that IRE1α also promotes myoblast fusion during overload-induced muscle growth (Fig. 8). Whole transcriptome analysis of IRE1α knockdown myoblasts showed gene enrichment of negative regulators of myoblast fusion. In addition, transcript levels of multiple profusion molecules are reduced in IRE1α knockdown cultured myoblasts and in the 5d-injured muscle of *Ern1^{scKO}* mice (Fig. 5). Furthermore, we demonstrate that IRE1α promotes myoblast fusion by increasing the levels of sXBP1, which regulates the gene expression of profusion molecules. This is evidenced by the findings that knockdown of XBP1 in cultured myoblasts also inhibits their fusion and results in the dysregulation of various molecules

involved in myoblast fusion. More importantly, forced expression of IRE1α or sXBP1 in cultured myoblasts is sufficient to induce expression of specific profusion molecules and facilitate myotube formation (Figs. 5, 6 and EV5).

Myomaker, a membrane protein, is critical for myoblast fusion. A recent study demonstrated that a promoter of *Mymk* contains E-Box motifs to which the MyoD transcription factor binds to induce the gene expression of myomaker for myotube formation (Zhang et al, 2020). The results of the present study suggest the IRE1α-XBP1 signaling axis promotes myoblast fusion through the upregulation of the *Mymk* gene (Figs. 6, 7). Transcriptionally active sXBP1 protein regulates gene expression of target molecules by interacting with the "ACGT" core-binding domain within their promoter or regulatory region. Several target genes regulated by sXBP1 have been identified that play important roles in ER proteostasis (Acosta-Alvear et al, 2007; Lin et al, 2021). We have identified two sXBP1 binding sites in the mouse *Mymk* gene and validated the enrichment of sXBP1 to these sites during myogenic differentiation. Furthermore, our loss-of-function and gain-of-function approaches demonstrate that IRE1α/XBP1 regulates gene expression of myomaker and that overexpression of myomaker is sufficient to rescue fusion defects in IRE1α knockdown myogenic cultures (Fig. 7). Interestingly, our RNA-Seq analysis also showed that knockdown of IRE1α in cultured myoblasts leads to the dysregulation of expression of a few other positive and negative regulators of myoblast fusion (Fig. 5D). However, the mechanisms by which IRE1α regulates the levels of those molecules remain unknown. Future studies will investigate whether IRE1α-XBP1 signaling modulates the gene expression of other fusion-related molecules in a similar fashion as myomaker during myogenesis.

Recently, He et al, showed that IRE1α signaling promotes muscle regeneration through RIDD-dependent decay of myostatin mRNA (He et al, 2021). In contrast, we found that IRE1α does not affect the mRNA levels of myostatin or other known RIDD substrates in regenerating skeletal muscle of mice (Roy et al, 2021). Instead, our RNA-Seq analysis of IRE1α knockdown cultures in this study showed a reduction in the mRNA levels of myostatin (Fig. EV4A). In addition, we did not find any significant difference in the early markers of myogenic differentiation, suggesting that IRE1α does not promote myoblast fusion through RIDD-dependent degradation of myostatin mRNA. Indeed, our results clearly show that IRE1α endonuclease activity causes the processing of XBP1 mRNA, which results in information on transcriptionally active XBP1 transcription factor, leading to enhanced gene expression of profusion molecules.

While our experiments suggest a role of IRE1α/XBP1 signaling in myoblast fusion, it remains unknown whether this pathway is activated due to ER stress during myogenesis or through signaling crosstalk with other pathways. Interestingly, we found that in addition to profusion molecules, the knockdown of IRE1α or XBP1 reduces gene expression of many molecules involved in the ER-associated degradation (ERAD) pathway (Figs. 5C and EV4). It is possible that the activation of UPR pathways is essential for myogenesis and one of the critical functions of the IRE1/XBP1 arm of the UPR is to augment myoblast fusion (Afroze and Kumar, 2019; Bohnert et al, 2018).

There are multiple reports suggesting that all three arms of the UPR are activated during myogenesis (Afroze and Kumar, 2019; Bohnert et al, 2018; Miyake et al, 2016; Tan et al, 2021; Xiong et al, 2017).

However, the mechanisms of activation of the UPR remain unknown. Since myogenic differentiation involves the synthesis of a new set of proteins, it is possible that an increase in the rate of protein synthesis during myogenesis can cause an accumulation of unfolded proteins which results in ER stress and activation of the UPR. Intriguingly, calcium has been shown to play a pivotal role in myoblast fusion during myogenesis (Abmayr and Pavlath, 2012; Hindi et al, 2013; Rochlin et al, 2010; Sampath et al, 2018). Disruption in calcium levels also causes stress in the ER, leading to the activation of the UPR (Hetz et al, 2015). Therefore, changes in calcium dynamics during myogenesis may also be another potential mechanism of activation of the UPR in myogenic cells.

Interestingly, our experiments showed that overexpression of IRE1α or sXBP1 in myoblasts does not induce fusion in growth conditions. Rather, myotube formation is improved when IRE1α or sXBP1-overexpressing myoblasts are incubated in a differentiation medium, suggesting that the activation of the UPR is a physiological response required for muscle formation and that distinct arm of the UPR regulates different aspects of myogenesis. While there is limited knowledge directly linking the ERAD pathway with myoblast fusion, our results provide initial evidence that IRE1α-mediated regulation of ERAD-associated molecules could also facilitate myoblast fusion during muscle regeneration.

In summary, our results suggest that the IRE1α/XBP1 signaling axis promotes myoblast fusion during regenerative myogenesis and overload-induced muscle growth in adult mice. While more investigation is needed to investigate mechanisms of action of IRE1α in myogenic cells, we provide initial evidence that augmenting the levels of IRE1α could improve muscle growth and repair in various muscle disorders.

# Methods

### Reagents and tools table

| Antibody | Source and catalog number | Dilution | Analysis |
|---|---|---|---|
| Polyclonal rabbit-anti-phospho-ERN1 (S724) | Abnova # PAB12435 | 1:1000 / 1:200 | WB / IF |
| Monoclonal rabbit-anti-IRE1alpha | Cell Signaling Technology # 3294 S | 1:1000 / 1:200 | WB / IF |
| Monoclonal rabbit-anti-XBP1s (E9V3E) | Cell Signaling Technology # 40435 | 1:1000 / 1:50 | WB / ChIP |
| Monoclonal rabbit-anti-GAPDH | Cell Signaling Technology # 2118 | 1:1000 | WB |
| Monoclonal mouse-anti-FLAG-M1 | Sigma # F-3040 | 1:500 | WB |
| Monoclonal rabbit-anti-phospho-SAPK/JNK (T183/Y185) | Cell Signaling Technology # 4668 S | 1:1000 | WB |
| Monoclonal rabbit-anti-SAPK/JNK | Cell Signaling Technology # 9252 S | 1:1000 | WB |
| Monoclonal rabbit-anti-phospho-p65 NF-κB | Cell Signaling Technology # 3033 | 1:1000 | WB |

| Antibody | Source and catalog number | Dilution | Analysis |
|---|---|---|---|
| Monoclonal rabbit-anti-total-p65 NF-κB | Cell Signaling Technology # 8242 | 1:1000 | WB |
| Polyclonal rabbit-anti-NF-κB p100/p52 | Cell Signaling Technology # 4882 | 1:1000 | WB |
| Polyclonal rabbit-anti-phospho-Glycogen synthase kinase-3 (GSK-3) | Cell Signaling Technology # 9336 | 1:1000 | WB |
| Monoclonal mouse-anti-Pax7 | DSHB # PAX7 | 1:200 / 1:100 | WB / IF |
| Monoclonal mouse-anti-Myosin heavy chain (embryonic) | DSHB # F1.652 | 1:200 / 1:15 | WB / IF |
| Monoclonal mouse-anti-MyoD | Santa Cruz Biotechnology # 377460 | 1:200 | WB / IF |
| Monoclonal mouse-anti-Myogenin | DSHB # F5D | 1:200 / 1:100 | WB / IF |
| Monoclonal mouse-anti-Myosin heavy chain | DSHB # MF20 | 1:250 / 1:100 | WB / IF |
| Polyclonal rabbit-anti-Laminin | Sigma # L9393 | 1:500 | IF |
| Goat anti-Mouse IgG1 Alexa Fluor 568 | Life Technologies # A21124 | 1:1000 | IF |
| Goat anti-Mouse IgG2 Alexa Fluor 594 | Life Technologies # A211135 | 1:1000 | IF |
| Goat anti-Rabbit IgG Alexa Fluor 488 | Life Technologies # 11034 | 1:1000 | IF |
| Goat anti-Mouse IgG2b Alexa Fluor 488 | Life Technologies # A21141 | 1:1000 | IF |
| **Oligonucleotides and sequence-based reagents** | | | |
| PCR/qRT-PCR primers | This study | Table EV1 | |

## Animals

Floxed *Ern1* (*Ern1*^fl/fl^) mice as described (Iwawaki et al, 2009) were crossed with *Pax7-CreERT2* mice (Jax strain: B6;129-Pax7^tm2.1(cre/ERT2)Fan^/J) to generate satellite cell-specific *Ern1*-knockout (i.e., *Ern1*^scKO^) mice. Myofiber-specific Ern1-KO (*Ern1*^mKO^) mice have been described previously (Roy et al, 2021). All mice were in the C57BL6 background. Mice were housed in individual cages in an environmentally controlled room (23 °C, 12-h light-dark cycle) with ad libitum access to food and water. The genotype of mice was determined by performing PCR from tail DNA. We used 9–16 weeks old male and female mice for our experimentation. For Cre-mediated inducible deletion of *Ern1* in satellite cells, the mice were given intraperitoneal (i.p.) injections of tamoxifen (10 mg per Kg body weight) in corn oil for four consecutive days and kept on tamoxifen-containing standard chow (Harlan Laboratories, Madison, WI) for the entire duration of the experiment. *Ern1*^fl/fl^ mice were also injected with tamoxifen following the same regimen and served as controls. All the animals were handled according to the approved institutional animal care and use committee (IACUC) protocol (PROTO201900043) of the University of Houston. All surgeries were performed under anesthesia, and every effort was made to minimize suffering.

## Skeletal muscle injury and in vivo fusion assay

TA muscle of adult mice was injected with 50 µl of 1.2% $BaCl_2$ (Sigma Chemical Co.) dissolved in saline to induce necrotic injury as described (Hindi and Kumar, 2016). The mice were euthanized at different time points after injury and TA muscle was isolated for biochemical and morphometric analysis. To study the fusion of muscle progenitor cells in vivo, the mice were given an intraperitoneal injection of EdU (4 µg per gram body weight) at day 3 post-$BaCl_2$-medited injury of TA muscle. On day 11 after the EdU injection, the mice were euthanized, and TA muscle was isolated and transverse sections were made. The TA muscle sections were subsequently immunostained with anti-Laminin for marking boundaries of myofibers and processed for the detection of EdU$^+$ nuclei. The EdU$^+$ nuclei on muscle sections were detected as instructed in the Click-iT EdU Alexa Fluor 488 Imaging Kit (Invitrogen). DAPI stain was used to identify nuclei. Finally, images were captured and the number of intramyofiber EdU$^+$ myonuclei per myofiber, percentage of 2 or more EdU$^+$ centrally nucleated fibers, and percentage of EdU$^+$ myonuclei/total DAPI$^+$ nuclei were evaluated using NIH ImageJ software. Three to four different sections from the mid-belly of each muscle were included for analysis.

## Synergistic ablation (SA) surgery

Bilateral SA surgery was performed in adult *Ern1*$^{fl/fl}$ and *Ern1*$^{scKO}$ mice following the same procedure as described (Hindi et al, 2018; Hindi et al, 2017). In brief, male mice were anesthetized using isoflurane and the soleus, and ~60% of the gastrocnemius muscles were surgically removed while ensuring that the neural and vascular supply remained intact and undamaged for the remaining plantaris muscle. A sham surgery was performed for controls following exactly the same procedures except that gastrocnemius and soleus muscles were not excised. After 2d, the mice were given an intraperitoneal injection of EdU (4 µg/g body weight). At 14d of sham or SA surgery, the plantaris muscle was isolated and transverse muscle sections made were immunostained with anti-laminin and for detection of EdU and nuclei. The number of intrafiber EdU$^+$ myonuclei/myofiber was quantified using NIH ImageJ software.

## Histology and morphometric analysis

Uninjured or injured TA muscle was isolated from mice and sectioned with a microtome cryostat. For the assessment of muscle morphology and quantification of myofiber cross-sectional area (CSA), 8-µm-thick transverse sections of TA muscle were stained with Hematoxylin and Eosin (H&E). The sections were examined under an Eclipse TE 2000-U microscope (Nikon, Tokyo, Japan). For quantitative analysis, the average CSA of myofibers was analyzed in H&E-stained TA muscle sections using NIH ImageJ software. For each muscle, the distribution of myofiber CSA was calculated by analyzing ~250 myofibers.

## Immunohistochemistry

For immunohistochemistry studies, frozen TA or plantaris muscle sections were fixed in acetone or 4% paraformaldehyde (PFA) in PBS, blocked in 2% bovine serum albumin in PBS for 1 h followed by incubation with anti-Pax7 or anti-eMyHC and anti-laminin in blocking solution at 4 °C overnight under humidified conditions. The sections were washed briefly with PBS before incubation with goat anti-mouse Alexa Fluor 594 and goat anti-rabbit Alexa Fluor 488 secondary antibody for 1 h at room temperature and then washed three times for 15 min with PBS. Nuclei were counterstained with DAPI. The slides were mounted using a fluorescence medium (Vector Laboratories) and visualized at room temperature on a Nikon Eclipse TE 2000-U microscope (Nikon), a digital camera (Nikon Digital Sight DS-Fi1), and NIS Elements BR 3.00 software (Nikon). Image levels were equally adjusted using Photoshop CS6 software (Adobe).

## Myoblast isolation, culturing, and fusion assay

Primary myoblasts were prepared from the hind limbs of 8-week-old male or female mice following a protocol as described (Hindi et al, 2017). To induce differentiation, the cells were incubated in a differentiation medium (DM; 2% horse serum in DMEM) for different time points. The cultures were then fixed with 4% PFA in PBS for 15 min at room temperature and permeabilized with 0.1% Triton X-100 in PBS for 10 min. Cells were blocked with 2% BSA in PBS and incubated with mouse-anti-MyHC (clone MF20) overnight at 4 °C. The cells were then incubated with a secondary antibody at room temperature for 1 h. Nuclei were counterstained with DAPI for 3 min. The stained cultures were photographed and analyzed using a fluorescent inverted microscope (Nikon Eclipse TE 2000-U), a digital camera (Digital Sight DS-Fi1), and Elements BR 3.00 software (Nikon). To measure fusion efficiency, more than one hundred MyHC$^+$ myotubes containing 2 or more nuclei were counted. The fusion index was defined as the number of DAPI-stained nuclei in MyHC$^+$ myotubes (containing 2 or more nuclei) divided by the total number of DAPI-stained nuclei. More than five field images were analyzed per experimental group. To measure the average diameter of myotubes, 100–120 myotubes per group were included. For consistency, diameters were measured at the midpoint along with the length of the MyHC$^+$ myotubes. Myotube diameter was measured using the NIH ImageJ software. The results presented are from four to five independent experiments. Image levels were equally adjusted using Photoshop CS6 software (Adobe).

## FACS

To detect p-IRE1α and IRE1α expression in cultured myoblasts, following the labeling with antibodies against CD45, CD31, Sca-1, Ter-119, and α7-integrin, the cells were fixed with 1% PFA (Sigma-Aldrich) and permeabilized using 0.2% Triton X-100 (Thermo Fisher Scientific). The cells were then incubated with anti-p-IRE1α or anti-IRE1α and Alexa 488 secondary antibody (Invitrogen). FACS analysis was performed on a C6 Accuri cytometer (BD Biosciences) equipped with three lasers. The output data was processed, and plots were prepared using FCS Express 4 RUO software (De Novo Software).

## Viral vectors

pBABE-Puro (Addgene, Plasmid #1764), pBABE-Puro-EGFP (Addgene, Plasmid #128041), and Flag_HsIRE1a_pBabePuro

(Addgene, Plasmid #54337) were purchased from Addgene (Watertown, MA). The pBX-Myomaker construct was provided by Dr. Douglas Millay of Cincinnati Children's Hospital Medical Center, Cincinnati, OH. Mouse sXBP1 cDNA was isolated and ligated at BamHI and SalI sites in the pBABE-Puro plasmid. The integrity of the cDNA was confirmed by performing DNA sequencing. For generation of retrovirus, about $5 \times 10^6$ Platinum-E packaging cells (Cell Biolabs, Inc, San Diego, CA) were transfected with 5 µg of Flag_HsIRE1a_pBabePuro, pBABE-Puro-sXBP1, or pBABE-Puro-EGFP using FuGENE-HD (Promega, Madison, WI, USA). After 24 h of transfection, the medium was replaced with 10% fetal bovine serum. Forty-eight hours after transfection, viral supernatants were collected, filtered through 0.45-micron filters, and then added to primary myoblasts in growth media containing 10 µg/ml polybrene. After two successive retroviral infections, cells were grown for 48 h and selected in the presence of 2 µg/ml puromycin.

The pLKO.1-mCherry-Puro plasmid was kindly provided by Dr. Renzhi Han of Ohio State University. The target siRNA sequence was identified using BLOCK-iT™ RNAi Designer online software (Life Technologies). At least 2–3 siRNA sequences for the IRE1α gene were tested for efficient knockdown of target mRNA. The shRNA oligonucleotides were synthesized to contain the sense strand of target sequences for mouse IRE1α (GCTAACGCCTACTCTGTATGT) short spacer (CTCGAG), and the reverse complement sequences followed by five thymidines as an RNA polymerase III transcriptional stop signal. Oligonucleotides were annealed and cloned into pLKO.1-mCherry-Puro with AgeI/EcoRI sites. The insertion of shRNA and cDNA sequences in the plasmids was confirmed by DNA sequencing. For generation of lentiviral particles, $5 \times 10^6$ HEK293T cells were co-transfected with 5 µg psPAX2 (Addgene, Plasmid # 12260), 5 µg pMD2.G (Addgene, Plasmid # 12259) and 10 µg of pLKO.1-mCherry-scrambled shRNA or pLKO.1-mCherry-IRE1α shRNA using FuGENE-HD reagent (Promega, Madison, WI, USA). After 24 h of transfection, the media was replaced with fresh media containing 10% fetal bovine serum. Lentiviral particles were collected 48 h after transfection and filtered through 0.45-micron filters. Primary myoblasts were transduced with lentiviral particles containing scrambled or IRE1α-shRNA in a growth medium containing 10 µg/ml polybrene. Cells were grown for 48 h and selected in the presence of 2 µg/ml puromycin.

## Myomaker promoter-reporter gene activity

Mouse Mymk promoter fragment (~2100 bp) was isolated by performing PCR using genomic DNA as a template. The sequence for forward primer was 5′-GGTACCGGTACCTGTCTCTTATGAC AAATGTCCTCT-3′ and the reverse primer sequence was 5′-GATATCCTCGAGACTGGTTTCTTTCCCCCAT-3′. PCR products were isolated and ligated in front of the firefly luciferase reporter in pGL4.20[Luc2/Puro] Vector (Promega). Primary myoblasts were transfected with myomaker promoter-reporter plasmid along with vector alone (pCMV5 plasmid) or pCMV5-Flag-XBP1s plasmid (Addgene, Plasmid #63680). Transfection efficiency was controlled by cotransfection of myoblasts with Renilla luciferase encoding plasmid pRL-TK (Promega). After 24 h of transfection, the cells were incubated in a differentiation medium for 24 h, and the cell lysates were made. Specimens were processed for luciferase expression using a dual luciferase assay system with

reporter lysis buffer per the manufacturer's instructions (Promega). Luciferase measurements were made using a luminometer Spectramax® i3x (Molecular Devices).

## RNA isolation and qRT-PCR

RNA isolation and qRT-PCR were performed following a standard protocol as described (Hindi et al, 2017). In brief, total RNA was extracted from uninjured and injured TA muscle of mice or cultured myoblasts using TRIzol reagent (Thermo Fisher Scientific) and RNeasy Mini Kit (Qiagen, Valencia, CA, USA) according to the manufacturer's protocols. First-strand cDNA for PCR analyses was made with a commercially available kit (iScript cDNA Synthesis Kit, Bio-Rad Laboratories). The quantification of mRNA expression was performed using the SYBR Green dye (Bio-Rad SsoAdvanced - Universal SYBR Green Supermix) method on a sequence detection system (CFX384 Touch Real-Time PCR Detection System - Bio-Rad Laboratories). The sequence of the primers is described in Table EV1. Data normalization was accomplished with the endogenous control (β-actin), and the normalized values were subjected to a $2^{-\Delta\Delta Ct}$ formula to calculate the fold change between control and experimental groups.

## Chromatin immunoprecipitation (ChIP) assay

Chromatin immunoprecipitation was performed using SimpleChIP enzymatic Chromatin IP kit (Cell Signaling Technology, Cat #9003) according to the manufacturer's suggested protocol. Briefly, primary myoblasts were incubated in DM for 24 h followed by crosslinking, purification of nuclei, and chromatin shearing by sonication (eight times, 20 s each). Sheared chromatin was incubated overnight with antibodies against sXBP1 (Cell Signaling Technology) or Histone H3 (Cell Signaling Technology) followed by incubation with Protein G magnetic beads. Normal Rabbit IgG (Cell Signaling Technology) was used as a negative control for the immunoprecipitation experiment. Magnetic beads were briefly washed, and chromatin DNA was eluted, reverse crosslinked, and purified. Purified DNA was analysed for enrichment of 100–150 bp sequences by quantitative real time-PCR (40 cycles) or semi-quantitative standard PCR using specific primers designed for binding sites in the regulatory regions of Mymk, Hspa5, Dnajb9, and mouse Rpl30 (positive control) genes. Triplicates were used from two sets of experiments for the standardization of qRT-PCR results. The fold change between negative control IgG and anti-sXBP1 groups was calculated using $2^{-\Delta\Delta Ct}$ formula. Primers used for PCR reactions are provided in Table EV1.

## Western blot

TA muscle of mice or cultured primary myoblasts were washed with PBS and homogenized in lysis buffer (50 mM Tris-Cl (pH 8.0), 200 mM NaCl, 50 mM NaF, 1 mM dithiothreitol, 1 mM sodium orthovanadate, 0.3% IGEPAL, and protease inhibitors). Approximately, 100 µg protein was resolved on each lane on 8–12% SDS-PAGE gel, transferred onto a nitrocellulose membrane, and probed using a specific primary antibody (described in Reagents and Tools Table). Bound antibodies were detected by secondary antibodies conjugated to horseradish peroxidase (Cell Signaling Technology). Signal detection was performed by an enhanced chemiluminescence detection reagent (Bio-Rad). Approximate molecular masses

were determined by comparison with the migration of prestained protein standards (Bio-Rad).

## RNA sequencing and data analysis

Total RNA from control, IRE1α or XBP1 knockdown cultures was extracted using TRIzol reagent (Thermo Fisher Scientific) using the RNeasy Mini Kit (Qiagen, Valencia, CA, USA) according to the manufacturer's protocols. The mRNA-seq library was prepared using poly (A)-tailed enriched mRNA at the UT Cancer Genomics Center using the KAPA mRNA HyperPrep Kit protocol (KK8581, Roche, Holding AG, Switzerland) and KAPA Unique Dual-indexed Adapter kit (KK8727, Roche). The Illumina NextSeq550 was used to produce 75 base paired-end mRNA-seq data at an average read depth of ~38 M reads/sample. RNA-seq fastq data were processed using CLC Genomics Workbench 20 (Qiagen). Illumina sequencing adapters were trimmed, and reads were aligned to the mouse reference genome Refseq GRCm39.105 from the Biomedical Genomics Analysis Plugin 20.0.1 (Qiagen). Normalization of RNA-seq data were performed using a trimmed mean of $M$ values. Genes with fold change (FC) ≥|1.5| (or $\log_2$FC ≥|0.5|) and FDR ≤0.05 were assigned as differentially expressed genes (DEGs) and represented in volcano plot using ggplot function in R software (v 4.2.2). Over-representation analysis was performed with a hypergeometric test using WebGestalt (v 0.4.3) (Benjamini and Hochberg, 1995). Gene Ontology (GO) Biological Processes associated with the upregulated and downregulated genes were identified with an FDR cutoff value of 0.05. Network enrichment analysis was performed using the Metascape Gene Annotation and Analysis tool (metascape.org) as described (Zhou et al, 2019). Heatmaps were generated by using heatmap.2 function (Gu and Hubschmann, 2022) using z-scores calculated based on transcripts per million (TPM) values. The average absolute TPM values for the control group are provided in Table EV2. TPM values were converted to log (TPM + 1) to handle zero values. Genes involved in specific pathways were manually selected for heatmap expression plots. All the raw data files can be found on the NCBI SRA repository using the accession code PRJNA1014488.

## Statistical analyses and experimental design

The sample size was calculated using power analysis methods for a priori determination based on the SEM and the effect size was previously obtained using the experimental procedures employed in the study. For animal studies, we calculated the minimal sample size for each group as eight animals. Considering a likely drop-off effect of 10%, we set the sample size of each group of six mice. For some experiments, three to four animals were sufficient to obtain statistically significant differences. Animals of the same sex and same age were employed to minimize physiological variability and to reduce s.d. from the mean. The exclusion criteria for animals were established in consultation with a veterinarian and experimental outcomes. In case of death, skin injury, ulceration, sickness, or weight loss of >10%, the animal was excluded from the analysis. Tissue samples were excluded in cases such as freeze artifacts on histological sections or failure in the extraction of RNA or protein of suitable quality and quantity. We included animals from different breeding cages by random allocation to the different experimental groups. All animal experiments were blinded using

number codes until the final data analyses were performed. Statistical tests were used as described in the Figure legends. Results are expressed as mean ± SEM. Statistical analyses used two-tailed Student's $t$-test or two-way ANOVA followed by Tukey's multiple comparison test to compare quantitative data populations with normal distribution and equal variance. A value of $p \leq 0.05$ was considered statistically significant unless otherwise specified.

## Data availability

All the raw data files for the RNA-Seq experiment can be found on the NCBI SRA repository using the accession code PRJNA1014488.

The source data of this paper are collected in the following database record: biostudies:S-SCDT-10_1038-S44319-024-00197-4.

## Peer review information

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

## Acknowledgements

This work was supported by National Institute of Health grants AR081487 and AR059810 to AK. We thank the technical support from the Cancer Prevention and Research Institute of Texas core for RNA-Seq experiment. We thank Dr. Douglas Millay of Cincinnati Children's Hospital Medical Center (Cincinnati, OH) for providing pBX-Myomaker construct. We also thank Dr. Pengpeng Bi of University of Georgia (Athens, GA) for providing myomaker antibody.

## Author contributions

**Aniket, S Joshi**: Data curation; Investigation; Methodology; Writing—original draft; Writing—review and editing. **Meiricris Tomaz da Silva**: Data curation; Investigation; Methodology; Writing—original draft; Writing—review and editing. **Anirban Roy**: Data curation; Formal analysis; Investigation; Methodology; Writing—review and editing. **Tatiana, E Koike**: Data curation; Investigation; Methodology. **Mingfu Wu**: Supervision; Methodology; Writing—review and editing. **Micah, B Castillo**: Formal analysis; Validation; Methodology. **Preethi, H Gunaratne**: Formal analysis; Supervision; Validation. **Yu Liu**: Supervision; Investigation; Methodology. **Takao Iwawaki**: Resources; Methodology. **Ashok Kumar**: Conceptualization; Supervision; Funding acquisition; Validation; Visualization; Project administration; Writing—review and editing.

Source data underlying figure panels in this paper may have individual authorship assigned. Where available, figure panel/source data authorship is listed in the following database record: biostudies:S-SCDT-10_1038-S44319-024-00197-4.

## Disclosure and competing interests statement

The authors declare no competing interests.

# Expanded View Figures

**A.**

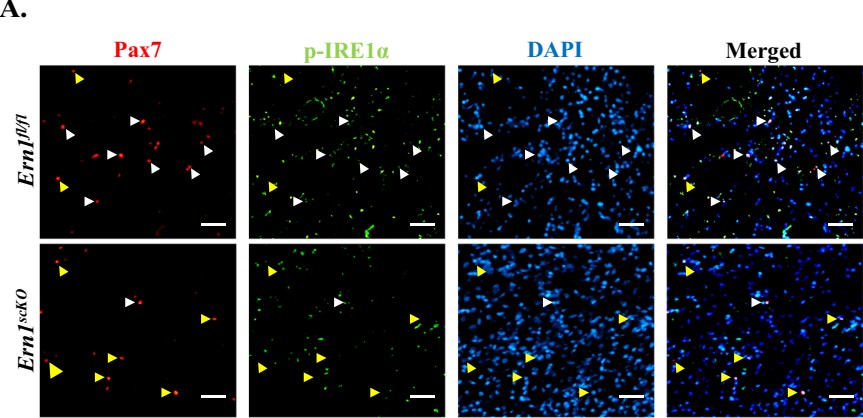

**B.**

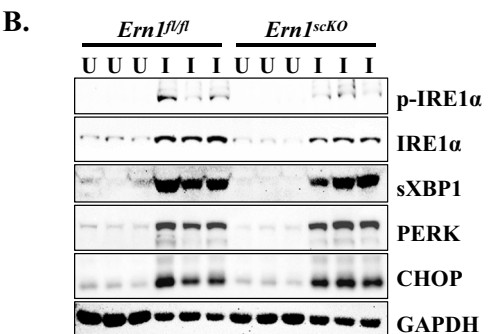

**Figure EV1. Deletion of IRE1α in satellite cells of Ern1scKO mice.**

(A) Representative photomicrographs of 5d-injured TA muscle of *Ern1fl/fl* and *Ern1scKO* mice after immunostaining for Pax7 and p-IRE1α protein and DAPI staining. White arrowheads point to Pax7 and p-IRE1α double-positive cells whereas yellow arrows point to Pax7 positive cells. Scale bar: 50 μm. (B) Immunoblots presented here demonstrate the levels of p-IRE1α, total IRE1α, sXBP1, PERK and CHOP protein in uninjured and 5d-injured TA muscle of *Ern1fl/fl* and *Ern1scKO* mice. U uninjured, I injured. Source data are available online for this figure.

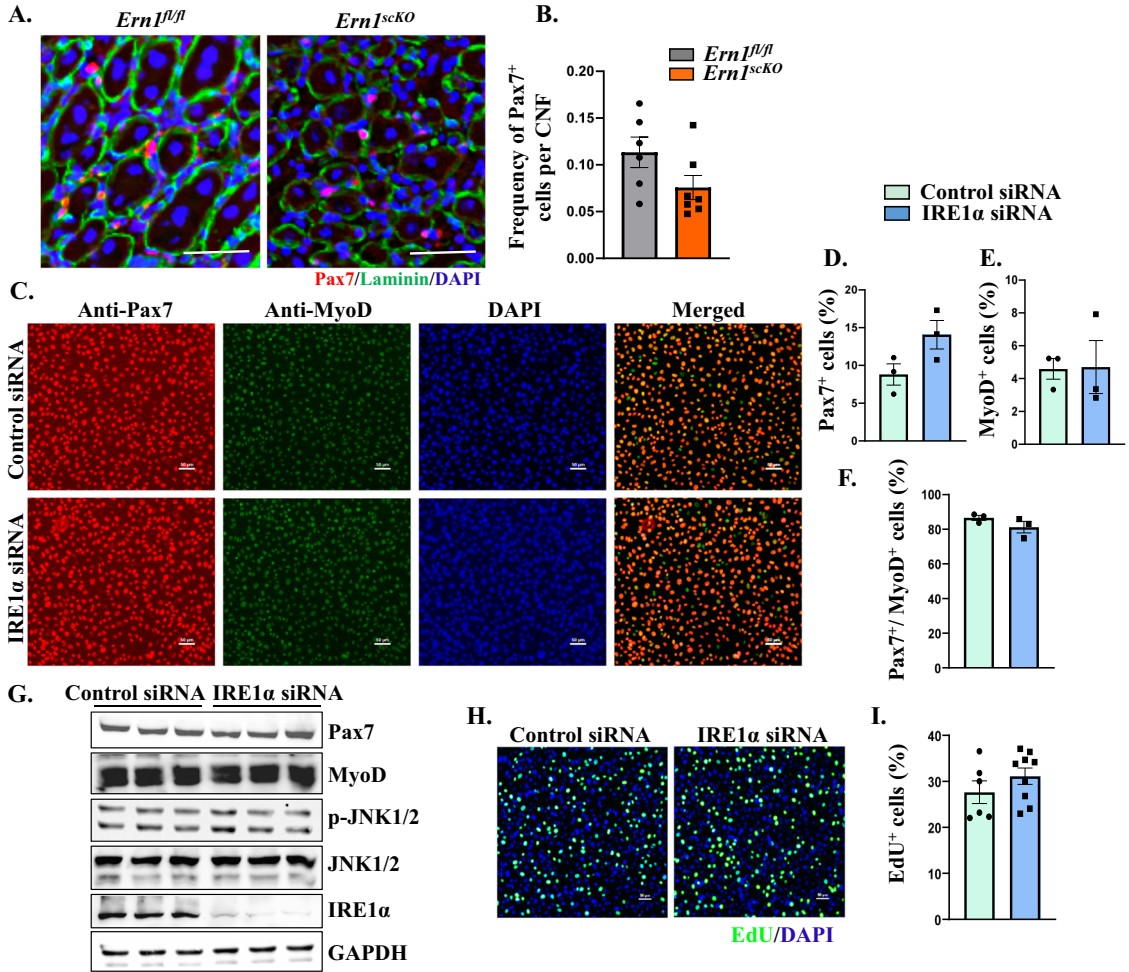

**Figure EV2. Targeted ablation of IRE1α does not affect the abundance and proliferation of satellite cells.**

(A) Representative images of 5d-injured TA muscle of *Ern1^fl/fl* and *Ern1^scKO* mice after immunostaining for Pax7 and Laminin protein and DAPI staining. Scale bar: 50 μm. (B) Quantification of the frequency of Pax7+ cells per centrally nucleated myofiber (CNF) in 5d-injured TA muscle of *Ern1^fl/fl* and *Ern1^scKO* mice. n = 6–7 mice per group. Data information: Data were presented as mean ± SEM. No significant difference was observed by unpaired Student t-test. (C) Representative photomicrographs of primary myoblast cultures transfected with control or IRE1α siRNA for 24 h followed by immunostaining for Pax7 and MyoD protein and DAPI staining. Scale bar: 50 μm. (D–F) Quantitative analysis of the proportion of (D) Pax7+, (E) MyoD+, and (F) Pax7+/MyoD+ cells in cultures transfected with control or IRE1α siRNA. n = 3 (biological replicates) per group. Data information: Data were presented as mean ± SEM. No significant difference was observed by unpaired Student t-test. (G) Immunoblots presented here show protein levels of Pax7, MyoD, p-JNK1/2, total JNK1/2, IRE1α, and an unrelated protein GAPDH in control and IRE1α siRNA transfected myoblast cultures. (H) Representative photomicrographs of EdU+/DAPI+ control and IRE1α knockdown cultures. Scale bar: 50 μm. (I) Quantification of the proportion of EdU+ cells in control and IRE1α knockdown myoblast cultures. n = 6–9 (biological replicates) per group. Data information: Data were presented as mean ± SEM. No significant difference was observed by unpaired Student t-test. Source data are available online for this figure.

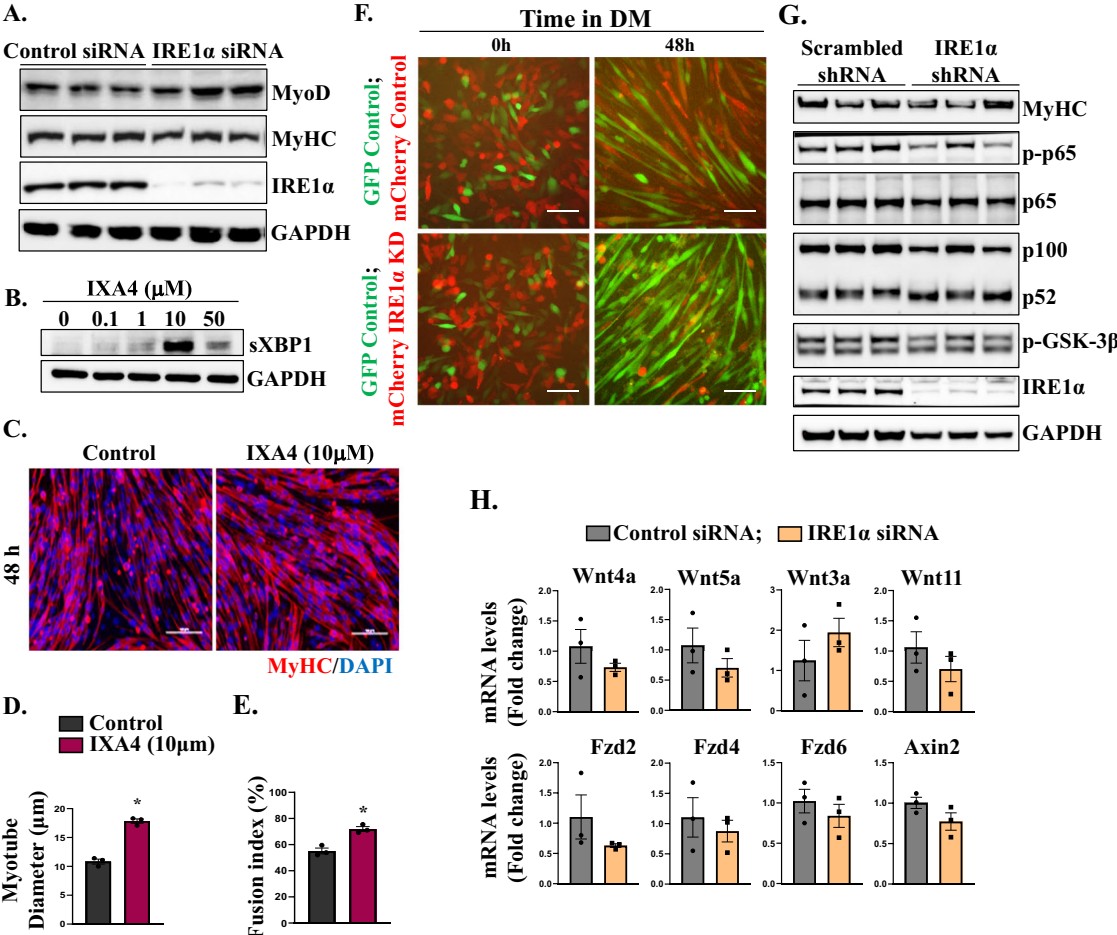

**Figure EV3. Knockdown of IRE1α inhibits myoblast fusion without affecting NF-κB and Wnt pathways.**

(A) Immunoblots presented here show levels of MyoD, MyHC, IRE1α, and GAPDH protein in myoblast cultures after 24 h of transfection with control siRNA or IRE1α siRNA. (B) The immunoblot presented here shows the levels of sXBP1 and GAPDH protein in primary myoblasts after 24 h of treatment with indicated concentrations (0, 0.1, 1, 10, 50 μM) of IXA4 compound. (C) Representative photomicrographs of cultures treated with vehicle alone (Control) or 10 μM IXA4 for 24 h followed by incubation in DM for 48 h and staining for MyHC. Scale bar: 50 μm. (D, E) Quantification of (D) average myotube diameter and (E) fusion index in control and IXA4-treated cultures at 48 h of incubation with DM. $n = 3$ (biological replicates) per group. Data information: Data were presented as mean ± SEM. *$p \leq 0.05$, values significantly different from control cultures analyzed by unpaired Student $t$-test. (F) Representative photomicrographs of co-cultured myogenic cells transduced with lentiviral particles expressing GFP protein (GFP Control), scrambled shRNA (Control), or IRE1α shRNA (IRE1 KD) along with mCherry protein at 0 and 48 h of addition of DM. Scale bar: 50 μm. (G) Immunoblots showing protein levels of MyHC, p-p65, p65, p100/p52, p-GSK-3β, and IRE1α in scrambled or IRE1 shRNA-expressing cultures at 24 h of differentiation. (H) Relative mRNA levels of *Wnt4a*, *Wnt5a*, *Wnt3a*, *Wnt11*, *Fzd2*, *Fzd4*, *Fzd6*, and *Axin2* in primary myoblasts transfected with control or IRE1α siRNA for 24 h. $n = 3$ (biological replicates) per group. Data information: Data were presented as mean ± SEM. No significance was observed by unpaired Student $t$-test. Source data are available online for this figure.

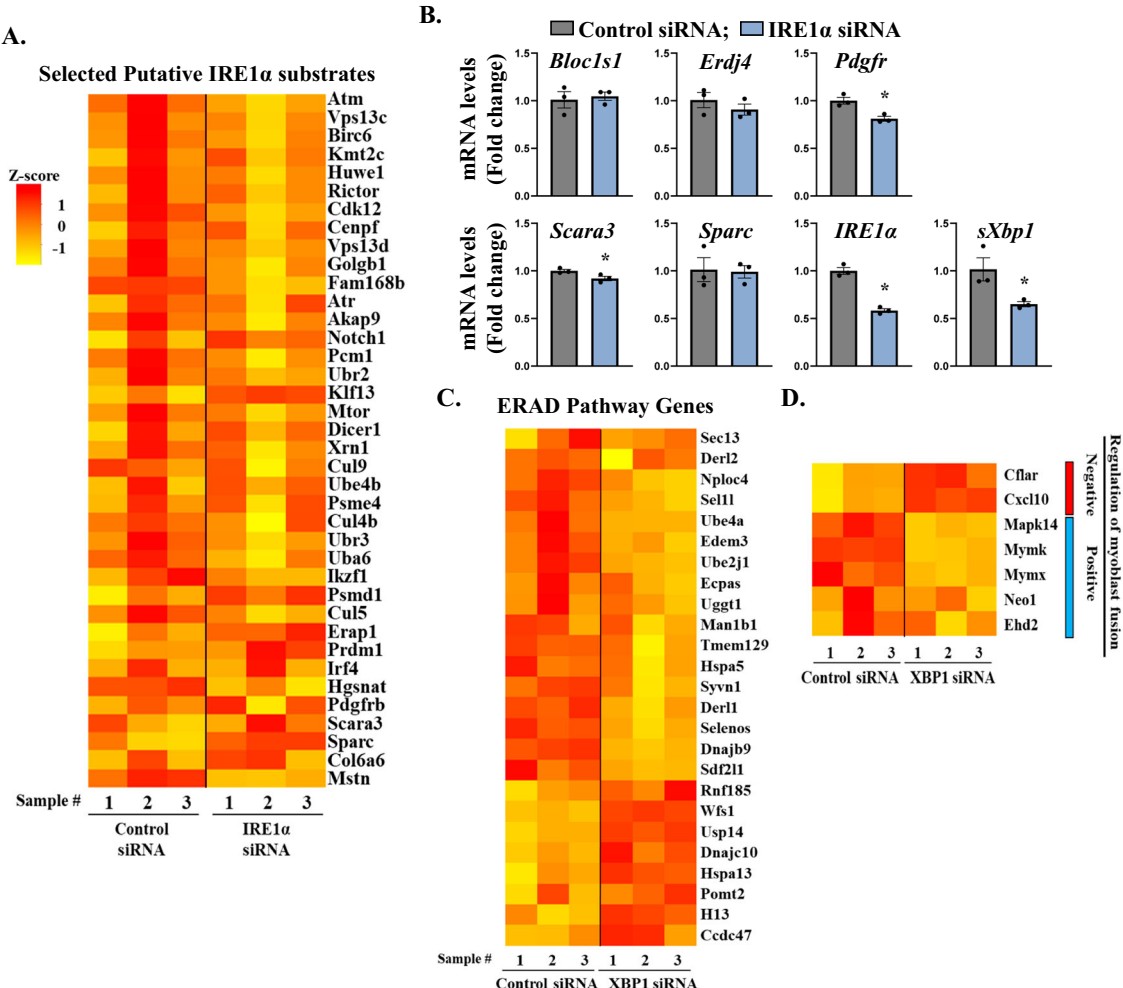

**Figure EV4. Gene expression analysis of IRE1α or XBP1 knockdown cultures.**

(A) Heatmap representing relative mRNA levels of selected putative RIDD substrates in IRE1α knockdown myoblast cultures at 24 h of addition of DM. (B) Relative mRNA levels of Bloc1s1, Erdj4, Pdgfr, Scara3, Sparc, IRE1α, and sXBP1 in control and IRE1α siRNA transfected myoblast cultures at 48 h of addition of DM. $n = 3$ (biological replicates) per group. Data information: Data were presented as mean ± SEM. *$p \leq 0.05$, values significantly different from cultures transfected with control siRNA analyzed by unpaired Student $t$-test. (C) Heatmap showing mRNA levels of molecules associated with ERAD pathway in XBP1 knockdown cultures compared to control cultures at 24 h of incubation in DM. (D) Heatmap showing mRNA levels of regulators of myoblast fusion in XBP1 knockdown cultures compared to control cultures at 24 h of incubation in DM. Source data are available online for this figure.

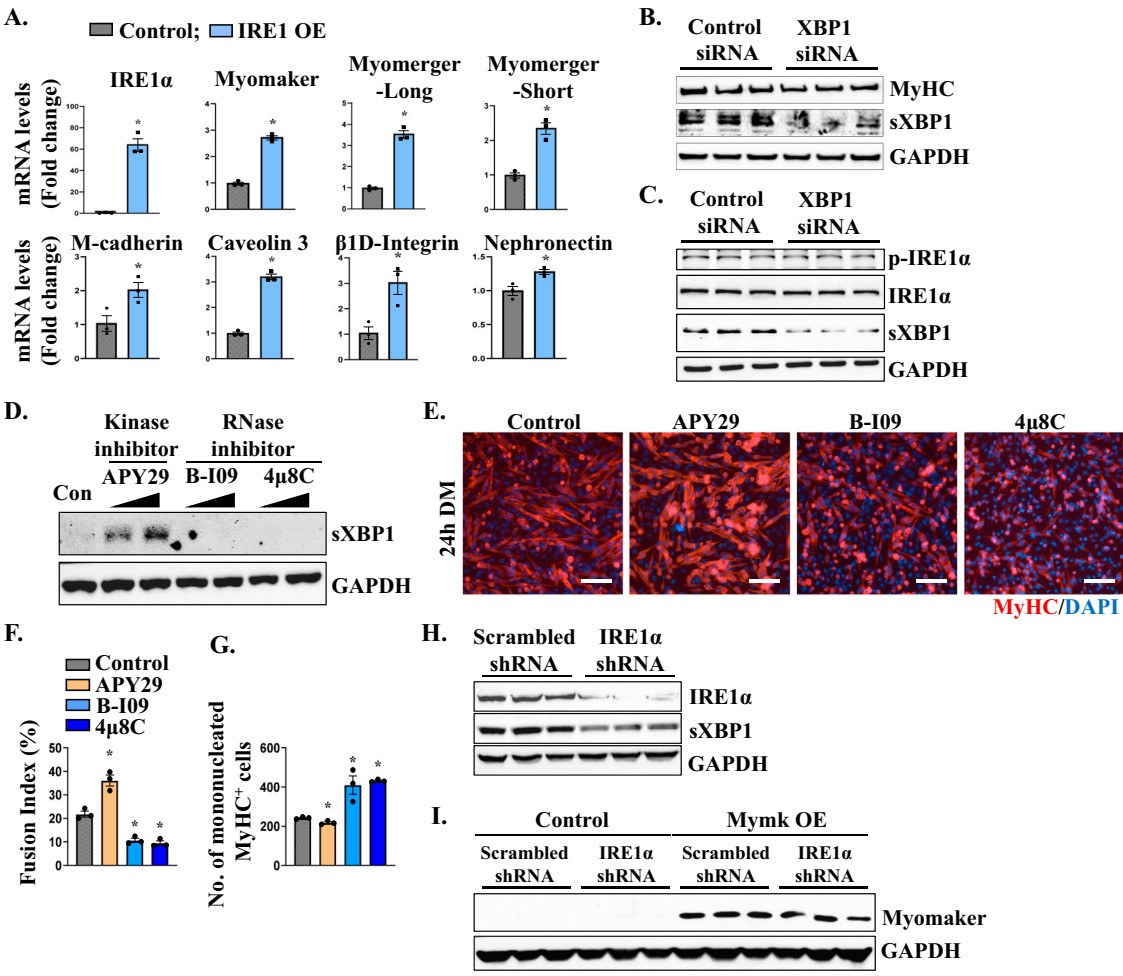

**Figure EV5. Effect of IRE1α overexpression or its pharmacological inhibition on myoblast fusion.**

(A) Relative mRNA levels of IRE1α, myomaker, myomerger-Long, myomerger-Short, M-cadherin, Caveolin-3, β1D-Integrin, and nephronectin in control and IRE1α overexpressing myoblast cultures. n = 3 (biological replicates) per group. Data information: Data are presented as mean ± SEM. *$p \leq 0.05$, values significantly different from control cultures analyzed by unpaired Student t-test. (B) Immunoblots presented here show the levels of MyHC, sXBP1, and GAPDH protein in control and XBP1 siRNA transfected cultures incubated in DM for 24 h. (C) Levels of p-IRE1α, IRE1α, sXBP1, and GAPDH protein in control and XBP1 knockdown cultures. (D) Levels of sXBP1 protein in control and APY29 (280 or 560 nM), B-I09 (1.23 or 2.46 μM), or 4μ8C (4 or 8 μM) treated primary myoblast cultures. (E) Representative photomicrographs of myoblast cultures treated with vehicle alone or with APY29 (280 nM), B-I09 (1.23 μM), or 4μ8C (4 μM) and incubated in DM for 24 h followed by immunostaining for MyHC protein and DAPI staining. Scale bar: 100 μm. (F, G) Quantification of (F) fusion index and (G) number of mononucleated MyHC⁺ cells in cultures. n = 3 (biological replicates) per group. Data information: Data were presented as mean ± SEM. *$p \leq 0.05$, values significantly different from control cultures analyzed by unpaired Student t-test. (H) Immunoblots showing knockdown of IRE1α and levels of sXBP1 and GAPDH in cultures expressing IRE1α shRNA. (I) The immunoblot presented here shows levels of myomaker and GAPDH protein in cultures transfected with *Mymk* cDNA. Source data are available online for this figure.

