## [Peer Review File · EMBO Reports]

The IRE1 α /XBP1 signaling axis drives myoblast fusion in adult skeletal muscle

Aniket Joshi, Meiricris Tomaz da Silva, Anirban Roy, Tatiana Koike, Mingfu Wu, Micah Castillo, Preethi Gunaratne, Yu Liu, Takao Iwawaki, and Ashok Kumar

Corresponding author(s): Ashok Kumar (akumar43@central.uh.edu)

Review Timeline:

Submission Date:	22nd Nov 23
Editorial Decision:	18th Jan 24
Revision Received:	16th Apr 24
Editorial Decision:	23rd May 24
Revision Received:	29th May 24
Accepted:	17th Jun 24

Editor: Deniz Senyilmaz Tiebe / Martina Rembold

Transaction Report:

Dear Ashok,

Thank you for the submission of your research manuscript to our journal, which was now seen by three referees, whose reports are copied below.

My apologies for this unusual delay in getting back to you. It took longer than anticipated to receive the full set of referee reports.

Referees express interest in the proposed role of the IRE1 α /XBP1 signaling in regulation of myoblast fusion in the muscle. However, they also raise significant concerns that need to be addressed to consider publication here.

Given these positive recommendations, we would like to invite you to submit a revised manuscript. Please revise your manuscript with the understanding that the referee concerns (as in their reports) must be fully addressed and their suggestions taken on board. Please address all referee concerns in a complete point-by-point response. Acceptance of the manuscript will depend on a positive outcome of a second round of review. It is EMBO reports policy to allow a single round of major experimental revision only and acceptance or rejection of the manuscript will therefore depend on the completeness of your responses included in the next, final version of the manuscript.

We realize that it is difficult to revise to a specific deadline. In the interest of protecting the conceptual advance provided by the work, we recommend a revision within 3 months. Please discuss the revision progress ahead of this time with me if you require more time to complete the revisions, or if you have questions or comments regarding the revision (also by video chat).

1. A data availability section providing access to data deposited in public databases is missing (where applicable).
2. Your manuscript contains statistics and error bars based on $n=2$. Please use scatter plots in these cases.

You can submit the revision either as a Scientific Report or as a Research Article. For Scientific Reports, the revised manuscript can contain up to 5 main figures and 5 Expanded View figures, and it should not exceed 27000 characters. If the revision leads to a manuscript with more than 5 main figures it will be published as a Research Article. In this case the Results and Discussion section should be separate. If a Scientific Report is submitted, these sections have to be combined. This will help to shorten the manuscript text by eliminating some redundancy that is inevitable when discussing the same experiments twice. In either case, all materials and methods should be included in the main manuscript file.

4) a .docx formatted letter INCLUDING the reviewers' reports and your detailed point-by-point responses to their comments. As part of the EMBO publication's Transparent Editorial Process, EMBO reports publishes online a Review Process File (RPF) to accompany accepted manuscripts. This File will be published in conjunction with your paper and will include the referee reports, your point-by-point response and all pertinent correspondence relating to the manuscript.

<https://www.embopress.org/page/journal/14693178/authorguide#transparentprocess>

5) a complete author checklist, which you can download from our author guidelines <https://www.embopress.org/page/journal/14693178/authorguide>. Please insert information in the checklist that is also reflected in the manuscript. The completed author checklist will also be part of the RPF.

6) Please note that all corresponding authors are required to supply an ORCID ID for their name upon submission of a revised manuscript (<<https://orcid.org/>>). Please find instructions on how to link your ORCID ID to your account in our manuscript tracking system in our Author guidelines <<https://www.embopress.org/page/journal/14693178/authorguide#authorshipguidelines>>

Additional information on source data and instruction on how to label the files are available: <https://www.embopress.org/page/journal/14693178/authorguide#sourcedata>

9) Our journal encourages inclusion of *data citations in the reference list* to directly cite datasets that were re-used and obtained from public databases. Data citations in the article text are distinct from normal bibliographical citations and should directly link to the database records from which the data can be accessed. In the main text, data citations are formatted as follows: "Data ref: Smith et al, 2001" or "Data ref: NCBI Sequence Read Archive PRJNA342805, 2017". In the Reference list, data citations must be labeled with "[DATASET]". A data reference must provide the database name, accession number/identifiers and a resolvable link to the landing page from which the data can be accessed at the end of the reference. Further instructions are available at <http://www.embopress.org/page/journal/14693178/authorguide#referencesformat>

- the name of the statistical test used to generate error bars and P values,
- the number (n) of independent experiments (please specify technical or biological replicates) underlying each data point,
- the nature of the bars and error bars (s.d., s.e.m.),
- If the data are obtained from n Program fragment delivered error `Can't locate object method "less" via package "than" (perhaps you forgot to load "than"?') at //ejpvfs23/sites23b/embor_www/letters/embor_decision_revise_and_review.txt line 56.' 2, use scatter blots showing the individual data points.

12) Please also note our reference format:

I look forward to seeing a revised version of your manuscript when it is ready. Please let me know if you have questions or comments regarding the revision.

Kind regards,

Deniz

Deniz Senyilmaz Tiebe, PhD
Scientific Editor
EMBO Reports

Referee #1:

The manuscript by Joshi et al. analyses the role of IRE1alpha/XBP1 signaling in myoblast fusion of adult skeletal muscle. It examines the outcomes both of satellite cell-specific ablation of IRE1alpha and of IRE1alpha silencing during muscle regeneration and myoblast fusion. Finally, it uncovers the role of sXBP1 and myomaker downstream of IRE1alpha in this process.

The findings are novel and interesting and are generally supported by solid experiments, with a few exceptions as detailed hereafter.

Figure 1. Quantifying the number of p-IRE1 α positive cells that colocalize with Pax7 positive cells would be useful to understand the percentage of satellite cells expressing p-IRE1 α in uninjured muscle. The same applies to the injured muscles.

Figure 2. Determination of IRE1alpha expression in the Ern-scKO satellite cells vs floxed controls, either at the mRNA or at the protein level, is mandatory.

Figure 3. In (H) IRE1 α expression should be shown to prove the efficacy of silencing.

Figure 5. mRNA level of genes reported in panel E should be measured in Ern-scKO satellite cells vs floxed controls.

Figure 6. Experimental evidence is required to show that sXBP1 o.e. rescues myomaker expression in IRE1alpha-silenced myoblasts.

Figure 7. To demonstrate that myomaker is directly transcribed by sXBP1 binding, it is necessary to perform transcription assays based on the expression of a reporter gene (e.g. luciferase or equivalent) driven by XBP1 binding sequences of the myomaker promoter upon sXBP1 o.e.

Typos:

Page 6: "We next investigated the effect of IRE1 α deletion in satellite cells on the level..." lacks a conclusion.

Figure 1. Title: "Activation of IRE1 α is muscle progenitor cells" should be "Activation of IRE1 α in muscle progenitor cells". (C) integrin instead of integrin.

Referee #2:

In this study the authors have expanded on their previous work to determine the role of the ER stress response protein IRE1a in skeletal muscle regeneration. They show that tamoxifen-induced Pax7-cre dependent knockout of IRE1a in adult mice impairs injury-induced muscle regeneration, revealing a muscle stem cell-autonomous function of IRE1a. Further analysis suggests that IRE1a KO may specifically inhibit myocyte fusion, which is corroborated by results of in vitro experiments. The authors have gone on to identify Myomaker as a direct target of transcriptional regulation by sXBP1, a downstream target of IRE1a. They conclude that the IRE1 -XBP1 pathway is a novel activator of myocyte fusion through the known fusion factor Myomaker. For the most part the study is well designed and executed, and the findings are of conceptual significance. However, how myocyte fusion is assessed in vivo as well as in vitro is not justified, and the following issues need to be addressed.

Major points:

1. It is curious that the nuclei in myofibers are EdU-positive even though they are supposed to be post-mitotic (Fig. 4 and Fig. 8). Regardless, why is the decreased number of EdU+ cells in IRE1a KO regenerating myofibers considered evidence for defective fusion?

2. Fusion index, a direct measurement of myocyte fusion, should be calculated for all the in vitro experiments (Fig. 4, Fig. 6, Fig. 7). Myotube diameter is not a good proxy for fusion as it is an indirect measurement of myotube size and it could also be influenced by fusion-independent hypertrophy. In addition, fusion index combined with myonuclei per myotube can potentially

reveal whether a defect is in the initial fusion or second-stage fusion (growth).

3. The eMyHC data in Fig. 3 (and Fig. 4A-C) need explanation in the Results section: why is "eMyHC cells per field" increased while the overall percentage of eMyHC-positive cells unchanged? What does each piece of the eMyHC data mean or imply?

4. Myomaker overexpression rescuing fusion (Fig. 7F-G) provides strong evidence beyond correlation that Myomaker mediates the function of IRE1 α in myoblast fusion. A similar experiment with overexpression of sXBP1 (resulting in rescue of Myomaker expression and myoblast fusion in IRE1 α KO or KD cells) would go a long way in substantiating the authors' conclusions.

Minor point:

5. It is not stated in the manuscript what ERN1 is or that it is the same gene as IRE1a.

Referee #3:

Previously the authors demonstrated that the myofiber-IRE1 and satellite cell-PERK UPR branches play important roles in preserving muscle regeneration capacity. Here the authors determined the necessity for satellite cell-IRE1 in skeletal muscle regeneration in adult mice. Joshi et al found that IRE1 and sXBP1 are activated in muscle progenitor cells. In contrast, levels of IRE1 and sXBP1 are reduced with myogenic differentiation. They further showed that the satellite cell IRE1-promoted myoblast fusion processes is required for muscle repair and growth. Mechanistically, Joshi et al. further revealed that sXBP1 activates gene expression of multiple profusion molecules including myomaker in differentiating myoblasts.

This manuscript is well written and presents a new role for the satellite cell IRE1-sXBP1 axis in the context of muscle regeneration. However, the authors have demonstrated that myofiber cell-specific IRE1-deficiency leads to impaired skeletal muscle regeneration in response to injury. Moreover, the authors have showed that PERK, another key UPR regulator, is important for satellite cell-mediated skeletal muscle regeneration. To advance the field of ER homeostasis-maintained muscle regeneration, the authors should have characterized the temporospatial- and cell type specific- profiles of key UPR regulators during progression and regression of muscle injury. For example, the authors could analyze levels/activities of key UPR markers in different cell populations within the TA muscle during course of muscle injury by western blot or flow cytometric analysis. Furthermore, the authors have showed that p-IRE1, IRE1 and XBP1 are induced in myoblasts and then decreased during differentiation. The causal mechanism(s) underlying these dynamic regulations should be further discussed. For example, what are the potential upstream regulators induce IRE1 activation? Another key missing molecular link is the direct action of IRE1 on XBP1 in myoblasts. Would sXBP1 OE rescue the IRE1 scKO-impaired regeneration capacity? Additionally, will treatments of an IRE1 kinase or RNase inhibitor have similar effects as genetic IRE1 deletion on myoblast function? The authors should provide solid evidence on this critical link.

Specific comments:

- Figure 1D, could the author elucidate the mechanism of a sustained sXBP1 expression post 24 hr of DM, wherein IRE1 was absent? Moreover, a tunicamycin or thapsigargin-treated XBP1 positive control should be included in the western blot analysis as an antibody control.
- Figure 2, it is critical to include expression levels of major UPR regulators, such as IRE1/XBP1/PERK/ATF6, in myoblasts and myofiber cells during injury time course in this experimental setting.
- Figure 3, p-IRE1, sXBP1 expression levels should be included in Fig. 3F&H.
- Figure 4, experiments with restoration of sXBP1 in the IRE1 siRNA groups or suppression of sXBP1 in the IRE1 OE groups should be included.
- Figure 5, the RNAseq was performed after 24 hr-differentiation induction, wherein there was a peak expression of XBP1 (Figure 1D). Did the authors find abolished sXBP1 expression in IRE1 siRNA samples at 24 hr of DM? Could the authors measure RIDD markers in IRE1 siRNA cells at later time points of differentiation, such as 48 and 72 hrs?
- Figure 6, it was demonstrated that XBP1 deletion leads to super activation of IRE1. Could the authors measure the p-IRE1 expression levels in the XBP1 siRNA groups?
- Figure 7, input of sXBP1 in CHIP assay should be included in Figure 7C&D.

RESPONSE TO REVIEWERS' COMMENTS

We are grateful to all the three reviewers for their time spent in reviewing our manuscript. We also appreciate valuable suggestions to improve the quality of our manuscript. We have now addressed all the comments by the reviewers. All the changes made in the manuscript in response to reviewers' comments are highlighted using yellow background. Our pointwise response to reviewers' comments is as follows:

REFEREE #1

The manuscript by Joshi et al. analyses the role of IRE1alpha/XBP1 signaling in myoblast fusion of adult skeletal muscle. It examines the outcomes both of satellite cell-specific ablation of IRE1alpha and of IRE1alpha silencing during muscle regeneration and myoblast fusion. Finally, it uncovers the role of sXBP1 and myomaker downstream of IRE1alpha in this process. The findings are novel and interesting and are generally supported by solid experiments, with a few exceptions as detailed hereafter.

Figure 1. Quantifying the number of p-IRE1 α positive cells that colocalize with Pax7 positive cells would be useful to understand the percentage of satellite cells expressing p-IRE1 α in uninjured muscle. The same applies to the injured muscles.

OUR RESPONSE: We have now provided the quantification of p-IRE1/Pax7 double positive cells in uninjured and injured muscle (new Figure 1B).

Figure 2. Determination of IRE1alpha expression in the Ern-scKO satellite cells vs floxed controls, either at the mRNA or at the protein level, is mandatory.

OUR RESPONSE: We have performed immunohistochemistry, which confirms the inactivation of IRE1 α protein in satellite cells of *Ern1^{scKO}* mice (Figure EV1A).

Figure 3. In (H) IRE1 α expression should be shown to prove the efficacy of silencing.

OUR RESPONSE: We have now added western blot data (now Figure 3G) showing that IRE1 α levels are reduced following transfection with IRE1 α siRNA.

Figure 5. mRNA level of genes reported in panel E should be measured in Ern-scKO satellite cells vs floxed controls.

OUR RESPONSE: We have now performed QRT-PCR, which showed that the levels of some of these molecules were also significantly reduced in 5d-injured TA muscle of *Ern1^{scKO}* mice compared to corresponding injured muscle of *Ern1^{fl/fl}* mice. These results are presented in new Figure 5F.

Figure 6. Experimental evidence is required to show that sXBP1 o.e. rescues myomaker expression in IRE1alpha-silenced myoblasts.

OUR RESPONSE: This is a really a good suggestion. We have performed this experiment and found that overexpression of sXBP1 improves myomaker levels in IRE1 α silenced myoblasts. Please refer to new Figures 7I and 7J.

Figure 7. To demonstrate that myomaker is directly transcribed by sXBP1 binding, it is necessary to perform transcription assays based on the expression of a reporter gene (e.g. luciferase or equivalent) driven by XBP1 binding sequences of the myomaker promoter upon sXBP1 o.e.

OUR RESPONSE: We have now performed the reporter gene expression analysis. Our results demonstrate that overexpression of sXBP1 significantly increases luciferase (reporter) activity in cultured myoblast incubated in DM (new Figure 7E).

Typos:

Page 6: "We next investigated the effect of IRE1 α deletion in satellite cells on the level..." lacks a conclusion.

Figure 1. Title: "Activation of IRE1 α is muscle progenitor cells" should be "Activation of IRE1 α in muscle progenitor cells". (C) integrin instead of integrin.

OUR RESPONSE: We have now corrected all these typos in the revised manuscript.

REFEREE #2

In this study, the authors have expanded on their previous work to determine the role of the ER stress response protein IRE1a in skeletal muscle regeneration. They show that tamoxifen-induced Pax7-cre dependent knockout of IRE1a in adult mice impairs injury-induced muscle regeneration, revealing a muscle stem cell-autonomous function of IRE1a. Further analysis suggests that IRE1a KO may specifically inhibit myocyte fusion, which is corroborated by results of in vitro experiments. The authors have gone on to identify Myomaker as a direct target of transcriptional regulation by sXBP1, a downstream target of IRE1a. They conclude that the IRE1 α -XBP1 pathway is a novel activator of myocyte fusion through the known fusion factor Myomaker. For the most part the study is well designed and executed, and the findings are of conceptual significance. However, how myocyte fusion is assessed in vivo as well as in vitro is not justified, and the following issues need to be addressed.

Major points:

1. It is curious that the nuclei in myofibers are EdU-positive even though they are supposed to be post-mitotic (Fig. 4 and Fig. 8). Regardless, why is the decreased number of EdU+ cells in IRE1a KO regenerating myofibers considered evidence for defective fusion?

OUR RESPONSE: The nuclei in uninjured myofibers are certainly post-mitotic. However, in response to myofiber injury, the satellite cells get activated which then undergo several rounds of proliferation, eventually differentiate, and fuse with injured myofibers. EdU (and BrdU as well) is a nucleoside analog of thymidine that is incorporated into DNA during S-phase of cell division/proliferation. EdU has been consistently used to pulse label the nuclei of proliferating cells including satellite cells/myoblasts *in vivo*. The EdU+ nuclei, which are observed in myofibers of regenerating muscle (14 days post injury), represent those nuclei, which came by fusion of muscle progenitor cells during muscle regeneration. A reduction in the number of EdU+ nuclei along with reduced myofiber CSA is an indication of reduced fusion during muscle regeneration. Indeed, this is a standard technique, which has been consistently used to study in vivo fusion of muscle progenitor cells with injured myofibers (e.g. PubMed ID: 28186492,

36400788, 29158520, 34633328 etc.).

2. Fusion index, a direct measurement of myocyte fusion, should be calculated for all the in vitro experiments (Fig. 4, Fig. 6, Fig. 7). Myotube diameter is not a good proxy for fusion as it is an indirect measurement of myotube size and it could be influenced by fusion-independent hypertrophy. In addition, fusion index combined with myonuclei per myotube can potentially reveal whether a defect is in the initial fusion or second-stage fusion (growth).

OUR RESPONSE: We completely agree with the reviewer. We have now added fusion index in Figs. 4L, 6J, and 7H and Figure EV3E. It was already there for in Figures 4F and 6C. Our results in Fig. 4H and 6D show increased number of mononucleated MyHC⁺ cells, which suggest that silencing of IRE1 or XBP1 inhibits primary myoblast fusion.

3. The eMyHC data in Fig. 3 (and Fig. 4A-C) need explanation in the Results section: why is "eMyHC cells per field" increased while the overall percentage of eMyHC-positive cells unchanged? What does each piece of the eMyHC data mean or imply?

OUR RESPONSE: We agree that this is bit confusing. When the myoblast fusion (not differentiation) is impaired, higher number eMyHC⁺ cells can accumulate per field of microscopy image due to their comparative small size. However, the overall circumference of the muscle is also reduced due to impairment of muscle regeneration. Thus, we believe measuring "eMyHC cells per field" is not a good measure of muscle formation. Therefore, we have removed this figure about "eMyHC cells per field" from the revised manuscript.

4. Myomaker overexpression rescuing fusion (Fig. 7F-G) provides strong evidence beyond correlation that Myomaker mediates the function of IRE1 α in myoblast fusion. A similar experiment with overexpression of sXBP1 (resulting in rescue of Myomaker expression and myoblast fusion in IRE1 α KO or KD cells) would go a long way in substantiating the authors' conclusions.

OUR RESPONSE: We have now performed this experiment. Our results demonstrate that overexpression of sXBP1 rescues Myomaker expression in IRE1 α KD cultures (new Figures 7I and J).

Minor point:

5. It is not stated in the manuscript what ERN1 is or that it is the same gene as IRE1 α .

OUR RESPONSE: This has been added in both the "Introduction" and the "Results" sections.

REFEREE #3

Previously the authors demonstrated that the myofiber-IRE1 and satellite cell-PERK UPR branches play important roles in preserving muscle regeneration capacity. Here the authors determined the necessity for satellite cell-IRE1 in skeletal muscle regeneration in adult mice. Joshi et al found that IRE1 and sXBP1 are activated in muscle progenitor cells. In contrast, levels of IRE1 and sXBP1 are reduced with myogenic differentiation. They further showed that the satellite cell IRE1-promoted myoblast fusion processes is required for muscle repair and growth. Mechanistically, Joshi et al. further revealed that sXBP1 activates gene expression of multiple

profusion molecules including myomaker in differentiating myoblasts.

This manuscript is well written and presents a new role for the satellite cell IRE1-sXBP1 axis in the context of muscle regeneration. However, the authors have demonstrated that myofiber cell-specific IRE1-deficiency leads to impaired skeletal muscle regeneration in response to injury. Moreover, the authors have showed that PERK, another key UPR regulator, is important for satellite cell-mediated skeletal muscle regeneration. To advance the field of ER homeostasis-maintained muscle regeneration, the authors should have characterized the temporospatial- and cell type specific- profiles of key UPR regulators during progression and regression of muscle injury. For example, the authors could analyze levels/activities of key UPR markers in different cell populations within the TA muscle during course of muscle injury by western blot or flow cytometric analysis.

OUR RESPONSE: These are interesting ideas where the activation of various components of ER stress can be studied in different cell types (using single cell or single nucleus RNA-Seq approaches) at various time points during muscle regeneration. We would definitely consider performing such experiments in future. The goal of the present study was to understand the cell autonomous role of the IRE1/XBP1 axis in satellite cells. Therefore, all our experiments are focused on characterizing the role and understanding the mechanisms of action of IRE1 α in muscle progenitor cells.

Furthermore, the authors have showed that p-IRE1, IRE1 and XBP1 are induced in myoblasts and then decreased during differentiation. The causal mechanism(s) underlying these dynamic regulations should be further discussed. For example, what are the potential upstream regulators induce IRE1 activation?

OUR RESPONSE: There are multiple reports suggesting the activation of the UPR during myogenic differentiation. There are also studies suggesting the PERK and ATF6 play important roles during myogenic differentiation. However, the mechanisms of activation of the UPR remains unknown. Since myogenic differentiation involves synthesis of new set of proteins, it is possible that an increase in rate of protein synthesis during myogenesis can cause accumulation of unfolded proteins which results in ER stress and activation of the UPR. Moreover, changes in Ca²⁺ dynamics during myogenesis may also be another potential mechanism of activation of the UPR in myogenic cells. We have now briefly discussed this aspect in the “Discussion” section of the manuscript (Page # 18, highlighted text).

Another key missing molecular link is the direct action of IRE1 on XBP1 in myoblasts. Would sXBP1 OE rescue the IRE1 scKO-impaired regeneration capacity? Additionally, will treatments of an IRE1 kinase or RNase inhibitor have similar effects as genetic IRE1 deletion on myoblast function? The authors should provide solid evidence on this critical link.

OUR RESPONSE: These are excellent suggestions. We have now performed new experiments, which demonstrate that overexpression of sXBP1 improves the levels of Myomaker expression in IRE1 α silenced myoblast cultures (Figure 7I and J). Furthermore, our new experiments demonstrate that pharmacological inhibitors of IRE1 kinase or RNase domains regulate myoblast fusion similar to IRE1 α or sXBP1 silencing (Figure EV5D-G).

Specific comments:

- Figure 1D, could the author elucidate the mechanism of a sustained sXBP1 expression post 24

hr of DM, wherein IRE1 was absent? Moreover, a tunicamycin or thapsigargin-treated XBP1 positive control should be included in the western blot analysis as an antibody control.

OUR RESPONSE: We do not observe sustained activation of sXBP1 during myogenesis. Indeed, the levels of sXBP1 in myoblasts are at peak at 24h after addition of DM and decreased thereafter. The sXBP1 antibody used in our manuscript is highly specific and has been validated using ER stressors and used for our other published studies as well (e.g. PMID: 34812145, 31138662, and 27206451)

• Figure 2, it is critical to include expression levels of major UPR regulators, such as IRE1/XBP1/PERK/ATF6, in myoblasts and myofiber cells during injury time course in this experimental setting.

OUR RESPONSE: Skeletal muscle injury leads to the accumulation cellular infiltrate that contains many different cell types (e.g. satellite cells, myofibers, immune cells, FAPs etc). We have deleted IRE1 α only in the satellite cells so it will be difficult to appreciate the contribution of satellite cell IRE1 α in the total levels of IRE1 α and other markers of the UPR in injured muscle. However, based on reviewer's suggestion, we performed western blots to analyze the levels of ER stress markers in uninjured and 5d-injured muscle. Our results showed that the levels of p-IRE1 α and total IRE1 α were considerably reduced in 5d-injured TA muscle of Ern1^{scKO} mice compared to corresponding 5d-injured TA muscle Ern1^{fl/fl} mice. There was also a trend towards reduced levels of sXBP1 in Ern1^{scKO} mice. However, the levels of PERK and CHOP protein remained comparable in the TA muscle of Ern1^{fl/fl} and Ern1^{scKO} mice (Figure EV1B). We could not detect the p-IRE1 α or sXBP1 protein in uninjured and 14d-injured TA muscle of mice potentially because the myogenesis has already completed at this time point.

• Figure 3, p-IRE1, sXBP1 expression levels should be included in Fig. 3F&H.

OUR RESPONSE: We have now included IRE1 α levels (now Figs. 3E and G) confirming the knockdown of IRE1 α . Similarly, we have data about pIRE1 α and total IRE1 α in Supplemental Figure EV1B. In addition, the levels sXBP1 protein in IRE1 α knockdown cultures are presented in Figure 6F and Figure EV5H.

• Figure 4, experiments with restoration of sXBP1 in the IRE1 siRNA groups or suppression of sXBP1 in the IRE1 OE groups should be included.

OUR RESPONSE: We have now included experiments in which sXBP1 was overexpressed in IRE1 α silenced cultures. Our experiments demonstrate that sXBP1 OE rescues the levels of myomaker in IRE1 α silenced cultures (Figure 7I and 7J)

• Figure 5, the RNAseq was performed after 24 hr-differentiation induction, wherein there was a peak expression of XBP1 (Figure 1D). Did the authors find abolished sXBP1 expression in IRE1 siRNA samples at 24 hr of DM?

OUR RESPONSE: Yes, we found drastic decrease in the levels of sXBP1 in IRE1 α siRNA cultures at 24h of DM (Figure 6F).

Could the authors measure RIDD markers in IRE1 siRNA cells at later time points of differentiation, such as 48 and 72 hrs?

OUR RESPONSE: We have now measured RIDD markers (e.g. Bloc1s1, Erdj4, Pdgr, Scara3, and Sparc) at 48 h. There was no significant difference in mRNA levels of Bloc1s1, Erdj4, and

Sparc whereas the levels of Scara3 and Pdgfr were reduced in control and IRE1 α KD cultures (Figure EV4B). These results suggest that IRE1 α RIDD activity may not be activated or involved during myogenesis.

- Figure 6, it was demonstrated that XBP1 deletion leads to super activation of IRE1. Could the authors measure the p-IRE1 expression levels in the XBP1 siRNA groups?

RESPONSE: We have now performed this experiment. However, we did not find any changes in levels of p-IRE1 or IRE1 protein in XBP1 knockdown cultures. These results are presented in Figure EV5C.

- Figure 7, input of sXBP1 in ChIP assay should be included in Figure 7C&D.

OUR RESPONSE: We have included 2% input as one of the samples in ChIP assay. Please refer to Figure 7C and 7D.

Dear Ashok,

Thank you for submitting your revised manuscript. It has now been seen by one of the original referees.

As you can see, the referees find that the study is significantly improved during revision and recommend publication. However, I need you to address the points below before I can accept the manuscript.

- Please address the remaining minor concerns of the referee #2.
- The maximum number of keywords we can accommodate is 5 for technical reasons. Thus, please remove two of the existing keywords.
- Please rename the Conflict of Interests section as "Disclosure Statement and Competing Interests".
- Please remove the Author Contributions section from the manuscript.
- We note a discrepancy in the spelling of the name of one of the authors - i.e. Preethi H. Gunaratne in the manuscript file vs. Preethi Guanaratne in the manuscript submission system.
- Please enter CPRIT RP180734 into the manuscript submission system in case it is a grant number relevant for the manuscript.
- Please convert Table EV2 into 'Reagents and Tools Table' (<https://www.embopress.org/page/journal/14693178/authorguide#textformat>) and leave it in the manuscript text. Please refer to Table EV1 in the 'Reagents and Tools Table' under the subheading 'Oligonucleotides and sequence-based reagents'. Please see <https://www.embopress.org/doi/full/10.15252/msb.20178071> for an example. Please update the callouts of the tables in the text and the names of the remaining EV Table accordingly.
- Please remove Table EV1 and Table EV3 and their legends from the manuscript text and provide them as separate files, which includes their legends.
- Please provide a link that directly resolves to the dataset PRJNA1014488 in the Data Availability section. Please remove the statement 'All relevant data related to this manuscript are available from the authors upon reasonable request.'
- We note the following regarding the Source Data:
 - o Please insert boxes on the microscopy images denoting the area represented in the figures.
 - o The source data files of microscopy images should be renamed in a way that they match the labeling in the figures.
 - o The source data for Figure 1C are cropped to the same extent as the figure panel itself.
- Our production/data editors have asked you to clarify several points in the figure legends:
 - o Please note that a separate 'Data Information' section is required in the legends of figures 2c-e, g-h; 4b-c, e-h, k-l; 5a, e-f; 6b-e, i-j; 8a, c, e; EV 2b, d-f, i; EV 3d-e, h; EV 4b; EV 5a, f-g.
 - o Please note that information related to n is missing in the legends of figures 2c-e; 4b-c; 6i-j; 8a, c.
 - o Although 'n' is provided, please describe the nature of entity for 'n' in the legends of figures 4k-l; 5e; 6b-e; 7d-e, g-h, j; 8d-f; EV 2i; EV 3d-e, h; EV 4b; EV 5a, f-g.
- Papers published in EMBO Reports include a 'synopsis' and 'bullet points' to further enhance discoverability. Both are displayed on the html version of the paper and are freely accessible to all readers. The synopsis includes a short standfirst summarizing the study in 1 or 2 sentences (max 35 words) that summarize the paper and are provided by the authors and streamlined by the handling editor. I would therefore ask you to include your synopsis blurb and 3-5 bullet points listing the key experimental findings.
- In addition, please provide an image for the synopsis. This image should provide a rapid overview of the question addressed in the study but still needs to be kept fairly modest since the image size cannot exceed 550 (width) x 300-600 (height) pixels.

Thank you again for giving us to consider your manuscript for EMBO Reports, I look forward to your minor revision.

Kind regards,

Deniz

--

Deniz Senyilmaz Tiebe, PhD
Editor
EMBO Reports

Referee #2:

The authors have done an admirable job addressing previous points with additional data. The only issue remaining for this reviewer is regarding fusion index: several new fusion index data have been added, but authors state that fusion index "was already there in Figures 4F and 6C", which is not accurate. The data in 4F and 6C shows the distribution of myotubes of various numbers of nuclei; it is not the same as fusion index which would reveal overall fusion capacity. For instance, a shift in % from myotubes containing 5-10 nuclei to myotubes containing 2-4 nuclei could be accompanied by an UNCHANGED total fusion

index, which would suggest a specific defect in the second-stage fusion. Although a reader could probably make an estimation of fusion index based on the data presented in 4F and 6C, I strongly suggest that fusion index be added as separate graphs for the images in Fig. 4D and 6A (48 hr alone would suffice).

A minor note: a typo pointed out by Reviewer 1 is still there - in the title of Figure 1 legend "is" should be "in".

RESPONSE TO REVIEWER'S COMMENTS**Referee #2**

Reviewer Comment # 1: The authors have done an admirable job addressing previous points with additional data. The only issue remaining for this reviewer is regarding fusion index: several new fusion index data have been added, but authors state that fusion index "was already there in Figures 4F and 6C", which is not accurate. The data in 4F and 6C shows the distribution of myotubes of various numbers of nuclei; it is not the same as fusion index, which would reveal overall fusion capacity. For instance, a shift in % from myotubes containing 5-10 nuclei to myotubes containing 2-4 nuclei could be accompanied by an UNCHANGED total fusion index, which would suggest a specific defect in the second-stage fusion. Although a reader could probably make an estimation of fusion index based on the data presented in 4F and 6C, I strongly suggest that fusion index be added as separate graphs for the images in Fig. 4D and 6A (48 hr alone would suffice).

OUR RESPONSE: We have now added fusion index at 48h. The fusion index results are in new Fig. 4I and 6F.

Reviewer comment # 2: A minor note: a typo pointed out by Reviewer 1 is still there - in the title of Figure 1 legend "is" should be "in".

OUR RESPONSE: This has now been corrected.

Prof. Ashok Kumar
University of Houston
Pharmacological and Pharmaceutical Sciences
4349 Martin Luther King Boulevard
Houston, TX 77204-5039
United States

Dear Prof. Kumar,

Since I am the secondary editor on your manuscript and my colleague Deniz is currently out of office, I have temporarily taken over the handling of your manuscript. I have checked all files and revisions and am very pleased to accept your manuscript for publication in the next available issue of EMBO reports. Thank you for your contribution to our journal.

With best regards,
